# Integrated β-catenin, BMP, PTEN, and Notch signalling patterns the nephron

**Nils O Lindström[1,2]\***, **Melanie L Lawrence[3]**, **Sally F Burn[4†]**, **Jeanette A Johansson[1†]**, **Elvira RM Bakker[5†]**, **Rachel A Ridgway[6†]**, **C-Hong Chang[3]**, **Michele J Karolak[7]**, **Leif Oxburgh[7]**, **Denis J Headon[1]**, **Owen J Sansom[8]**, **Ron Smits[5]**, **Jamie A Davies[3]**, **Peter Hohenstein[1,2]\***

[1]Division of Developmental Biology, The Roslin Institute, University of Edinburgh, Easter Bush, United Kingdom; [2]MRC Human Genetics Unit, MRC Institute of Genetics and Molecular Medicine, Edinburgh, United Kingdom; [3]Centre for Integrated Physiology, University of Edinburgh, Edinburgh, United Kingdom; [4]Department of Genetics and Development, Columbia University, New York, United States; [5]Laboratory of Gastroenterology and Hepatology, Erasmus MC, University Medical Centre, Rotterdam, Netherlands; [6]Department of Invasion and Metastasis, Cancer Research UK Beatson Institute, Glasgow, United Kingdom; [7]Center for Molecular Medicine, Maine Medical Center Research Institute, Scarborough, United States; [8]Beatston Institute for Cancer Research, Glasgow, United Kingdom

**Abstract** The different segments of the nephron and glomerulus in the kidney balance the processes of water homeostasis, solute recovery, blood filtration, and metabolite excretion. When segment function is disrupted, a range of pathological features are presented. Little is known about nephron patterning during embryogenesis. In this study, we demonstrate that the early nephron is patterned by a gradient in β-catenin activity along the axis of the nephron tubule. By modifying β-catenin activity, we force cells within nephrons to differentiate according to the imposed β-catenin activity level, thereby causing spatial shifts in nephron segments. The β-catenin signalling gradient interacts with the BMP pathway which, through PTEN/PI3K/AKT signalling, antagonises β-catenin activity and promotes segment identities associated with low β-catenin activity. β-catenin activity and PI3K signalling also integrate with Notch signalling to control segmentation: modulating β-catenin activity or PI3K rescues segment identities normally lost by inhibition of Notch. Our data therefore identifies a molecular network for nephron patterning.

**\*For correspondence:** nils.lindstrom@roslin.ed.ac.uk (NOL); peter.hohenstein@roslin.ed.ac.uk (PH)

†These authors contributed equally to this work

**Competing interests:** The authors declare that no competing interests exist.

**Reviewing editor**: Elaine Fuchs, Rockefeller University, United States

## Introduction

All adult vertebrates depend on renal nephrons accurately performing a diverse set of roles that protect and maintain blood homeostasis. The diversity of roles that the nephron performs is reflected in the range of symptoms of abnormal nephron function and the consequent diseases range from, for example, Bartter syndrome (abnormal water and salt loss), acidosis, and rickets to essential hypertension (*Simon et al., 1996*; *Schedl, 2007*). In spite of the importance of proper nephron function and segmentation, little is known regarding how the nephron patterns and how different specialised segments form during nephrogenesis. Thus far, only a handful of genes have been shown to control the development of specific nephron segments of which none have been explicitly connected to an explanation for how patterning is regulated along the whole length of the nephron.

Nephrons form during embryonic development, when Wnt9b, secreted by the ureteric bud, activates a canonical β-catenin-mediated pathway in a population of overlying Six2[+] mesenchymal nephron progenitor-cells (*Kobayashi et al., 2008*; *Karner et al., 2011*; *Park et al., 2012*; *Das et al., 2013*). In

**eLife digest** The main function of the kidney is to filter blood to remove waste and regulate the amount of water and salt in the body. Structures in the kidney—called nephrons—do much of this work and blood is filtered in a part of each nephron called the glomerulus. The substances filtered out of the blood move into a series of 'tubules', another part of the nephrons, from where water and soluble substances are reabsorbed or excreted as the body requires. If the nephrons do not work correctly, it can lead to a wide range of health problems—from abnormal water and salt loss to dangerously high blood pressure.

For organs and tissues to develop in an embryo, signalling pathways help cells to communicate with each other. These pathways control what type of cells the embryonic cells become and also help neighbouring cells work together to form specialised structures with particular functions. Much is unknown about how the nephron develops, including how its different structures coordinate their development with each other so that they form in the right position in the nephron.

A protein called beta-catenin was already known to play an important role in the signalling pathways that trigger the earliest stages of nephron formation. Lindström et al. further investigated how this protein helps the nephron to develop by using a wide range of techniques, including growing genetically altered mouse kidneys in culture and capturing images of the developing nephrons with time-lapse microscopy. The combined results reveal that the levels of beta-catenin activity coordinate the development of the different structures in the nephron. The beta-catenin protein is not equally active in all parts of the nephron; instead, it forms a gradient of different activity levels. The highest levels of beta-catenin activity occur in the tubules at the furthest end of the developing nephron; this activity gradually decreases along the length of the nephron, and the glomerulus itself lacks beta-catenin activity altogether. Experimentally manipulating the levels of beta-catenin at different points along the nephron caused those cells to take on the wrong identity, causing parts of the nephron to form in the wrong place.

Lindström et al. were also able to establish that the signalling pathway controlled by beta-catenin activity interacts with three other well-known signalling pathways as part of a network that controls nephron development. More research is required to find out which signal activates beta-catenin in the first place and from where in the kidney this signal comes. It also remains to be discovered how a particular cell in the tubule interprets the exact activities of the different signals to give the cell its specific identity for that place in the nephron. A better understanding of these sorts of processes will eventually help build new kidneys for people with kidney failure.

the canonical WNT pathway, WNT signalling results in the destabilisation of the GSK-3β/CK1α/AXIN2/APC complex and prevents the normal tagging of cytosolic β-catenin for degradation. Stabilised β-catenin translocates to the nucleus and together with members of the TCF family of transcription factors controls the expression of a wide range of target genes (*Clevers and Nusse, 2012*). One of these β-catenin target genes, *Wnt4*, triggers a mesenchymal-to-epithelial transition (MET) (*Stark et al., 1994*; *Kispert et al., 1998*) in the nephron progenitor cells and these reconfigure into an epithelial renal vesicle (RV). Following the MET, the RV becomes polarised and connects to the ureteric bud at its distal end (*Georgas et al., 2009*), and during subsequent developmental steps, the nephron starts to display additional signs of pattern-formation along its proximal–distal axis. Several distinct cell-populations form and these produce the different segments of the adult nephron (*Saxen, 1987*); a Wt1⁺ cell population gives rise to proximal structures, a Jag1⁺ population to the medial part and Lgr5⁺ cells generate the distal nephron segments (*Armstrong et al., 1993*; *Cheng et al., 2003*; *Chen and Al-Awqati, 2005*; *Cheng et al., 2007*; *Kreidberg, 2010*; *Barker et al., 2012*). These segments are in turn further subdivided into functionally specialised portions, which express specific combinations of transmembrane transporters/channels for salts, glucose, and metals (*Raciti et al., 2008*). How the differentiation of these segments is regulated remains unknown. The initiation of the nephron MET is driven by β-catenin signalling (*Kobayashi et al., 2008*; *Karner et al., 2011*; *Park et al., 2012*), but the Wnt4 driven MET is most likely mediated by the non-canonical Ca²⁺–NFAT pathway (*Burn et al., 2011*; *Tanigawa et al., 2011*). It remains uncertain by what mechanism and at what precise stage the Six2⁺ cells or the RV develop distinct nephron segment lineages (*Lindström et al., 2013*). Post-MET,

Wnt9b acts via the planar cell polarity pathway and controls the orientation of cell division and the elongation of collecting tubules (*Karner et al., 2009*). Wnt7b also has a role as it controls the development of the medulla and papilla of the kidney (*Yu et al., 2009*). Notch signalling has previously been identified as being important for the formation of the proximal tubule (*Cheng et al., 2003*, *2007*). *Notch2⁻/⁻* nephrons form no proximal tubules or glomeruli (*Cheng et al., 2007*). However, ectopic expression of the intracellular and active Notch1-domain (N1ICD) in nephrons blocks glomerular development (*Cheng et al., 2003*, *2007*; *Boyle et al., 2011*). N1ICD expression in Six2⁺ cells can actually substitute for Wnt9b and trigger nephron induction and MET (*Boyle et al., 2011*). Whether Notch or Wnt is important for the initial patterning of the nephron immediately post-MET remains to be determined.

Using in vivo and ex vivo techniques we demonstrate that a gradient of β-catenin activity, along the proximal–distal nephron axis, controls the differentiation of segment-specific cell fates. We further investigate how β-catenin activity is prevented in the proximal and medial segments and show that BMP/PTEN/PI3K signalling in the medial nephron actively promotes the medial segment identity whilst blocking β-catenin activity. In addition, we show that modulating β-catenin or PI3K activity partially rescues the nephron segment defect phenotypes associated with the loss of Notch function. Our findings provide a model where multiple signalling pathways are integrated to control nephron segment-identity specification.

## Results

### A β-catenin activity gradient is generated along the nephron axis

Regulation of β-catenin activity is essential for nephron induction and MET (*Davies and Garrod, 1995*; *Kuure et al., 2007*; *Park et al., 2007*). To determine whether β-catenin is involved in post-MET stages of nephron development, we tracked its activity in embryonic kidney organ cultures using a β-catenin signalling reporter mouse strain (*TCF/Lef::H2B-GFP*; *Ferrer-Vaquer et al., 2010*). Confocal and time-lapse microscopy indicated strong activity of the reporter in the ureteric bud as described before (*Ferrer-Vaquer et al., 2010*; *Burn et al., 2011*). Importantly, we also detected different GFP intensities, reporting β-catenin activity, along the proximal–distal axis of the nephron. The nomenclature we use to describe the domains of the proximal–distal axis is as defined by the GenitoUrinary Development Molecular Anatomy Project (gudmap.org) for S-shaped body nephrons. Confocal z-projections (*Figures 1A*, 3D rendering *Video 1*) of whole nephrons at an early stage of development show the signal being highest at the distal end of the nephron, where it connects to the ureteric bud, and gradually decreasing towards the proximal end. Time-lapse imaging of developing *TCF/Lef::H2B-GFP* expressing nephrons showed that the different GFP signal intensities propagated in a distal-to-proximal direction over time alongside the normal nephron growth and segmentation (*Figure 1— figure supplement 1A* and *Video 2*). Confocal imaging confirmed different GFP intensities in nephrons at later stages: S-shaped bodies (*Figure 1B* and *Figure 1—figure supplement 1B*) and more mature nephrons (data not shown), and we consistently found that the podocytes and their precursors at the extreme proximal end of the nephrons were almost completely devoid of β-catenin activity (*Figure 1A,B*, *Figure 1—figure supplement 1B*; *Video 1*). We quantified the *TCF/Lef::H2B-GFP* signal in cells located in the distal, medial, and proximal segments of nephrons and plotted their intensities against their position. The segments were defined with antibodies for Jag1 (medial segment; *Chen and Al-Awqati, 2005*; *Georgas et al., 2009*), Cdh1 (distal segment; *Cho et al., 1998*), and by morphology. The *TCF/Lef::H2B-GFP* signal intensities showed an exponentially decreasing gradient ($R2 = 0.999$; $n = 11$ nephrons) suggestive of a single source and first-order decay of the activating signal (*Figure 1C*). Jag1 immunofluorescent intensity data were used to indicate positions within nephrons. The GFP intensities measured in the distal segments differed from those in proximal segments by a minimum of a 15-fold difference to a maximum of a 72-fold difference (mean = 39-fold difference; $n = 10$ nephrons; *Figure 1C*).

To further assess the observed gradient in β-catenin activity, we analysed the expression and activity of the β-catenin protein directly. Using pan-β-catenin antibodies, we found β-catenin to be expressed throughout the developing nephrons (*Figure 1D*). Antibodies for β-catenin that was already tagged for degradation (phosphorylated at Ser33/Ser37/Thr41), most strongly labelled the proximal domain (twofold higher compared to medial and distal, $p = 6.2 \times 10^{-5}$ and $1.2 \times 10^{-4}$) (*Figure 1—figure supplement 1C*). Target genes of β-catenin (*Lef1* and *Ccnd1*) were also expressed

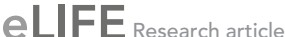

**Figure 1**. β-catenin activity levels form a reversed gradient along the axis of the nephron. (**A–B**) *TCF/Lef::H2B-GFP* expression in nephrons: (**A**) late renal vesicle/early comma-shaped body nephron, (**B**) S-shaped body nephron, lines: white–nephron axis, purple–ureteric bud, green–distal nephron, red–medial nephron, blue–proximal nephron/glomerular precursors. Heat-maps display signal intensity in different nephron segments. (**C**) Quantification of nuclear H2B-GFP and cell-membrane Jag1 antibody stain signal-intensity along the proximal–distal axis. Error bars represent SEM of pixels representative of 10 μm segments. Right-hand side graph shows mean values for segments, as identified by H2B-GFP and Jag1 profiles (n = 11 nephrons), error bars indicate SEM. p-values derived from t-tests. (**D**) Antibody stains against total β-catenin and phosphorylated β-catenin in S-shaped body nephron—Jag1 marking the medial segment. White dashed line indicating nephron axis.

The following figure supplement is available for figure 1:

**Figure supplement 1**. β-catenin reporter and antibody data show different β-catenin activity levels along the axis of the nephron.

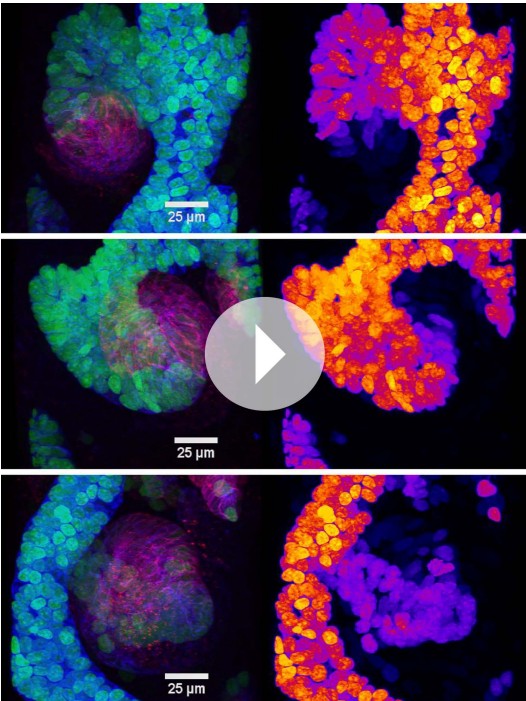

**Video 1**. 3D reconstructions of nephrons. 3D reconstruction of renal vesicle (top), S-shaped body (middle), and more mature nephron (bottom). Nephrons are positive for *TCF/Lef::H2B-GFP*, Jag1-red, Cdh1-blue (left panel) and the *TCF/Lef::H2B-GFP* reporter is shown in a heat-map overlay (right panel).

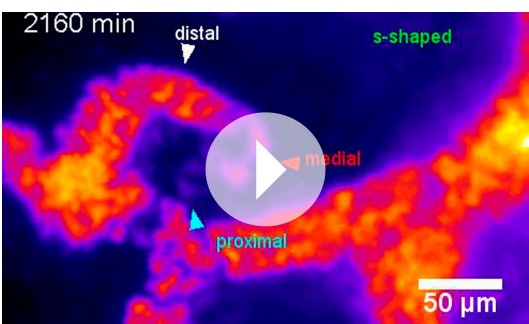

**Video 2**. Time-lapse capture of nephron. The nephron is the same as shown in *Figure 1—figure supplement 1A* (from a *TCF/Lef::H2B-GFP* reporter kidney) is shown during the earliest stages of nephron development. Segments and stages are annotated based on the brightfield channel (not shown).

along the gradient (*Figure 1—figure supplement 1D–F*). The distribution of β-catenin protein and the expression of direct β-catenin target genes support the existence of the activity gradient.

## Proximal and distal positional identities are controlled by different β-catenin activity levels

To investigate whether the β-catenin activity gradient has functional implications for the development of the nephron proximal–distal axis, we used inhibitors of GSK-3β (CHIR99021) and Tankyrase (IWR1) to positively and negatively modify β-catenin signalling ex vivo in whole organ cultures. We extensively characterised these small molecule inhibitors to ensure that they had their expected effect on the β-catenin signalling pathway; these data can be found in *Figure 2—figure supplements 1–4* and *Videos 3–4* and are briefly described here. First, we used the β-catenin reporter *TCF/Lef::H2B-GFP* and qRT-PCR analyses and confirmed that the inhibitors acted as expected (*Figure 2—figure supplement 1*). Second, because maximal activation of β-catenin has previously been suggested to be incompatible with MET (*Kuure et al., 2007*; *Park et al., 2007*, *2012*), we identified CHIR concentrations that induced nephron differentiation (PAX2 and PAX8 expression) but still permitted epithelialisation (CDH1 expression; *Figure 2—figure supplement 2*). Increased β-catenin signalling induced ectopic nephrons to form at the periphery of the kidneys and large nephron structures developed within the ureteric tree (*Figure 2—figure supplements 1,2*). Third, we demonstrated that these ectopic structures, just like the tree-bound structures, were derived from Six2[+] nephron progenitors by fate-mapping these using a *Six2-CreGFP* (*Dolt et al., 2013*) and a *Rosa26[tdRFP]* Cre reporter mouse model (*Luche et al., 2007*) (*Figure 2—figure supplement 3* and *Video 3*). Fourth, since pharmacological inhibitors can have multiple targets, we confirmed their specificity by combining activators and inhibitors of the WNT-pathway. *Pax8*-Cre lineage tracing and immunofluorescent analyses allowed us to show that by blocking β-catenin signalling downstream of GSK-3β (by administering ICG001 which binds CBP and prevents β-catenin/CBP interaction), but not upstream (IWR1), CHIR-induced ectopic nephron formation was inhibited (*Figure 2—figure supplement 4A* and *Video 4*). Fifth, additional inhibitors against components of the β-catenin/Canonical WNT-pathway also confirmed our findings: (BIO (GSK3β inhibitor) induced effects similar to CHIR; salinomycin (LRP6 inhibitor) induced effects similar to IWR1 (*Gandhirajan et al., 2010*; *Lu et al., 2011*); *Figure 2—figure supplement 4B*).

Using these inhibitors we modulated β-catenin signalling in *Wt1[+/GFP]* nephrons, where GFP labels the proximal cell population, allowing maturing podocytes within glomeruli to be recognized by their

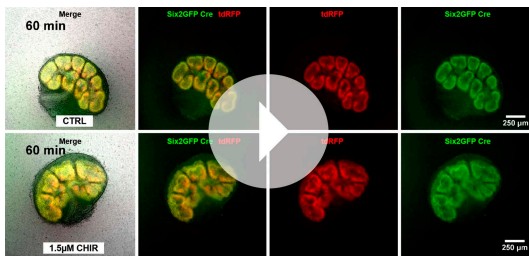

**Video 3**. Time-lapse capture of *Six2*^GFPCre^/Rosa26^tdRFP^ kidneys. Kidneys cultured in control medium and CHIR medium. Timing and conditions as shown in videos.

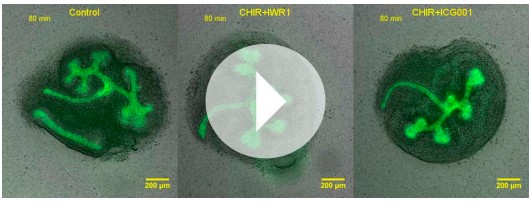

**Video 4**. Time-lapse capture of Pax8^Cre^ YFP^lox-stop^ kidneys. Kidneys cultured in control medium, IWR1 and CHIR medium, or ICG001 and CHIR medium. Timing and conditions as shown in videos.

bright concentrated GFP signal (***Figure 2A***, ***Video 5***). The *Wt1*^+/GFP^ also labels the nephron progenitor cells and the whole of the pre-tubular aggregate during the MET, but at lower intensities, in accordance with the known expression pattern of *Wt1* (***Kreidberg, 2010***). Time-lapse analysis showed that, by decreasing β-catenin activity, we favoured the differentiation of the proximal cell identity as was seen by an increase in maturation rate of glomeruli; in contrast, increased β-catenin activity blocked proximal identity formation and the formation of glomeruli (***Figure 2A–C***). Podxl antibody staining (***Figure 2D***), qRT-PCR analysis of additional markers for proximal differentiation (*Nphs2*, *Synpo*, *Nphs1*, *Podxl*), and *Wt1*^+/GFP^ reporter kidneys (***Figure 2—figure supplement 5A,B***) confirmed that these markers were expressed earlier when β-catenin activity was decreased and that they were repressed when β-catenin activity was increased. We consistently noticed a small number of individual cells in CHIR conditions that remained positive for Wt1 or Podxl (***Figure 2A,D***). Releasing these cells from increased β-catenin activity, by removing CHIR, allowed them to resume their development (***Figure 2E***). The proximal identity almost fully recovered and glomerular structures formed at almost the same size as those found in controls, show that the ectopically increased β-catenin activity had been actively suppressing the proximal identity (***Figure 2—figure supplement 5C***). In contrast, removing IWR1 did not result in the reversal of the phenotype (***Figure 2—figure supplement 5D***).

Distal cells express the epithelial stem-cell marker and β-catenin target *Lgr5* (***Barker et al., 2012***). To test whether the β-catenin gradient also controls the formation of the distal identity, we monitored *Lgr5-EGFP* expression in kidneys at different β-catenin activity levels using the *Lgr5*^+/EGFP-IRES-CreERT2^ mouse model (***Barker et al., 2012***). *Lgr5* was expressed as previously described (***Barker et al., 2012***), although we primarily detected strong *Lgr5-EGFP* expression in a subsection of the distal domain adjacent to the medial segment (***Figure 3A***). *Lgr5* expression levels were increased at high levels of β-catenin activity. When β-catenin activity was decreased, the expression domain extended longer although the actual GFP signal was at lower levels compared to controls and samples where β-catenin activity was increased (***Figure 3A***). We analysed the dynamics of *Lgr5* expression under different β-catenin activity levels using time-lapse imaging of *Lgr5*^+/EGFP-IRES-CreERT2^ kidneys. The number of *Lgr5* positive nephrons increased significantly at higher β-catenin activity levels (3.0×; p = 0.05) compared to controls or kidneys with decreased β-catenin activity (***Figure 3B,C***). However, in samples with decreased β-catenin activity, *Lgr5* positive nephrons emerged at an earlier time-point and these nephrons again appeared to be more elongated, but the GFP signal was lower compared to controls (***Figure 3B*** and ***Video 6***). Compared to controls, the actual number of *Lgr5* positive nephrons remained unchanged in IWR1 treated samples (***Figure 3C***). *Lgr5* is also an R-spondin receptor and mediates β-catenin signalling (***de Lau et al., 2011***). To test whether *Lgr5* was functionally modulating the β-catenin signalling gradient, and thereby controlling nephron patterning, we intercrossed the *Lgr5*^+/EGFP-IRES-CreERT2^ knockin mice to homozygosity and analysed these for segmentation defects. Homozygous mutants displayed no obvious phenotype in the developing kidney (***Figure 3—figure supplement 1*** and ***Video 7***).

Collectively, these data confirm that the cell populations along the proximal–distal axis of the nephrons are responsive to changes in β-catenin signalling and respond positively and negatively as would be predicted from the proposed β-catenin activity gradient.

**A**

**Proximal progenitor (Wt1$^{GFP/+}$) time-lapse 0 - 4000 min**

**B**

**Reduced β-cat activity increases glomerular maturation rate**

**C**

**Increased β-cat activity decreases glomerular differentiation**

**D**

CTRL    2µM IWR1    1.5µM CHIR

**E**

CTRL (120 hrs)    1.5µM CHIR (120 hrs)    1.5µM CHIR (48 hrs) + CTRL ( 72 hrs)

**Figure 2**. Pharmacological modulation of β-catenin signalling alters proximal segment development. (**A**) Time-lapse analysis of treated *Wt1$^{+/GFP}$* kidneys—same as shown in *Video 5*. (**B**) Quantification of mean time taken for first glomeruli to mature to crescent-shaped stage where glomeruli are tightly packed and exhibit a bright signal. (**C**) Mean number of mature glomeruli after 3800 min of culture. (**D**) Kidneys stained for podocyte marker Podxl, β-catenin target Lef1, and epithelial marker Cdh1. Arrowheads indicating structures positive for Podxl. (**E**) The proximal identity resumed its formation when CHIR was removed after 48 hr—white arrowheads indicate larger Podxl positive structures, blue arrowheads indicate very small Podxl positive structures, dashed line separates ectopic nephron zone (e.n.z.) nephrons from those inside the ureteric tree (UB).

The following figure supplements are available for figure 2:

**Figure supplement 1**. β-catenin signalling is altered in response to pharmacological inhibitors.

*Figure 2. Continued on next page*

*Figure 2. Continued*

**Figure supplement 2**. 'Just-right' β-catenin signalling levels drive MET.

**Figure supplement 3**. Ectopic nephrons form from Six2 expressing progenitors.

**Figure supplement 4**. Co-inhibition experiments confirm specificity of pharmacological inhibitors.

**Figure supplement 5**. The proximal cell-identity is promoted by decreased β-catenin signalling.

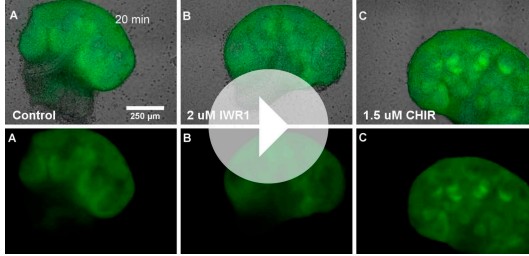

**Video 5**. Time-lapse capture of Wt1$^{+/GFP}$ kidneys. Kidneys cultured in control conditions or treated with IWR1 or CHIR. Timing, conditions, and scale as specified.

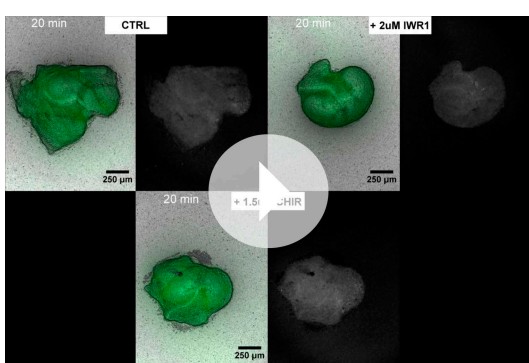

**Video 6**. Time-lapse capture of *Lgr5$^{+/EGFP-IRES-CreERT2}$* kidneys. Kidneys cultured in control conditions or treated with IWR1 or CHIR. Timing, conditions, and scale as specified.

## Modulating β-catenin activity shifts positional identities along the nephron

Having shown that the proximal and distal cell populations are both affected by increased β-catenin activity and the proximal is affected by decreased β-catenin activity, we hypothesized that the different levels of β-catenin signalling within the gradient specify the positions of identities in the nephron. If so, modulating β-catenin signalling might impose abnormal distal or proximal identities along the axis of the developing nephron (*Figure 4A*). We immunostained inhibitor treated nephrons with markers for proximal, medial, and distal identities (Wt1, Jag1, Cdh1) and measured the size of the domains in which each was expressed (*Figure 4B*). Increasing β-catenin did mildly increase total nephron lengths (1.17×, p = 0.017) compared to controls (*Figure 4C*), but the distal segment was significantly longer (1.8×, p = 0.0003) and the proximal segment was severely reduced (0.2×, p = 1.5 × 10$^{-6}$). The medial segment showed no significant change in length (1.3×, p = 0.067). These data confirm that both the distal and proximal nephron segments respond to increased β-catenin signalling according to our hypothesis, but the medial remained unchanged in size. Decreasing β-catenin signalling, on the other hand, resulted in much elongated nephrons (*Figure 4B,C*: 1.7×, p = 1.8 × 10$^{-14}$), as we had previously noticed (*Figure 3A*). Here, the length of the proximal segment was unchanged compared to controls (1.0×, p = 0.93), but as we showed above, this segment was more mature in appearance and gene expression (*Figure 2*). The elongation of the nephron was primarily due to the increases in both the distal (2.5×, p = 0.0002) and the medial segments (1.7×, p = 0.0007) and the nephrons appeared thinner. Although nephron identities responded to increased β-catenin as our model predicted, the large morphological changes obscured any subtle changes in segmentation in those nephrons with decreased β-catenin activity. To address this, we tested whether a gradual increase in β-catenin activity, which only mildly shifted activity levels away from the normal, would give a gradual decrease in proximal identity as would be expected if identities are β-catenin dosage-dependent. This was observed in nephrons that we exposed to different incremental concentrations of CHIR (*Figure 4—figure supplement 1*). qRT-PCR analysis of RNA from whole treated kidneys confirmed that a mild increase in β-catenin activity promoted expression of most distal segment genes (*Wnt4, Pax8, Fgf8, Lhx1, Lgr5*) (*Figure 4D*) thus mirroring the effect of decreasing β-catenin activity; surprisingly, *Pax2* did not respond similarly to *Pax8*. Medial segment control genes (*Jag1, Dll1, Heyl,* and *Irx2*) did not increase in response

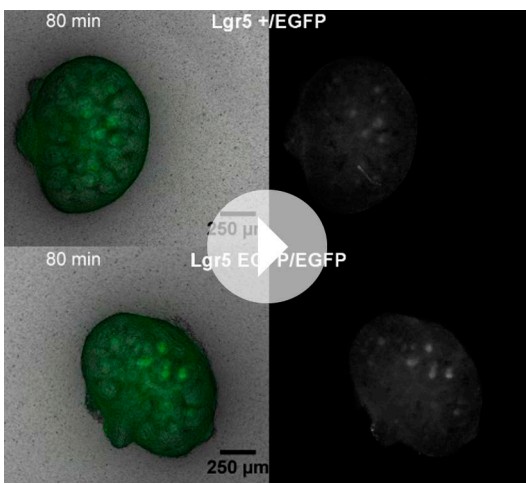

**Video 7.** Time-lapse capture of *Lgr5*[+/EGFP-IRES-CreERT2] and *Lgr5*[EGFP-IRES-CreERT2/EGFP-IRES-CreERT2] kidneys. Kidneys cultured in control conditions over 6 days. Timing and scale as specified.

to CHIR as distal genes did. At the highest β-catenin activity levels used [6 μM], nephron formation was reduced (*Figure 2—figure supplements 1–2*) and the expression profile thus dropped as would be expected (*Figure 4D*). To test if the observed phenotypes were due to a direct effect on the nephrogenic lineage or an indirect effect via the ureteric bud, we isolated metanephric mesenchyme away from the ureteric bud, induced it to form nephrons with spinal cord (*Grobstein, 1953*, *1955*), and modulated β-catenin activity therein. Comparable to the phenotype observed in whole kidney rudiments, we found that inhibition of β-catenin activity favoured patterning towards the proximal fate, whereas increasing its activity had the opposite effect (*Figure 4E*).

Previous genetic approaches to investigate the role of β-catenin during nephron formation have been hampered by the problem that when β-catenin is maximally activated in nephron progenitors MET is concurrently blocked (*Kuure et al., 2007*; *Park et al., 2007*). Our data, however, suggest that MET is only incompatible with very high levels of β-catenin activity (*Figure 4D* and *Figure 2—figure supplements 1–2*), therefore identifying a dose–response correlation between β-catenin activity and renal MET. To confirm this, and to test if nephron segmentation defects could also be generated by genetic means, we made use of *Apc* and *Ctnnb1* (the gene encoding β-catenin) mutants that only result in mild increases in β-catenin activity. The *Ctnnb1*[E654] model (*van Veelen et al., 2011*) expresses a Y654E mutation in the endogenous β-catenin gene (*Ctnnb1*) leading to an increased propensity to signal, whereas the *Apc*[1572T] and *Apc*[1638N] models (*Fodde et al., 1994*; *Gaspar et al., 2009*) carry endogenous *Apc*-truncations associated with low or moderate activation of β-catenin signalling, respectively. Such hypomorphic alleles by themselves or in combinations can provide a series of increasing β-catenin activity levels (*Kielman et al., 2002*; *Buchert et al., 2010*). We generated different combinations of these alleles that display increasing levels of β-catenin activity in order of increasing β-catenin activity: *Ctnnb1*[+/E654], *Ctnnb1*[E654/E654], *Apc*[+/1572T] *Ctnnb1*[+/E654], and *Apc*[+/1638N] *Ctnnb1*[+/E654] (*Gaspar et al., 2009*; *van Veelen et al., 2011*). We cultured E12.5 kidney rudiments for 5 days and stained them for different compartments of the kidneys with antibodies against Wt1, Jag1, Podxl, Lam, and Cdh1. *Ctnnb1*[+/E654] and *Ctnnb1*[E654/E654] kidneys (*Figure 4—figure supplement 2*) were indistinguishable from wild-type (*Ctnnb1*[+/+]) kidneys (data not shown). Kidneys from *Apc*[+/1572T] *Ctnnb1*[+/+] and *Apc*[+/1572] *Ctnnb1*[+/E654] embryos had normal nephrons (*Figure 4—figure supplement 2*), though some kidneys developed supernumerary ureteric buds (data not shown). The mutants with the highest level of β-catenin activity, *Apc*[+/1638N] *Ctnnb1*[+/E654] displayed the most severe phenotype. The kidneys from such embryos showed effects ranging from those with severely reduced and morphologically abnormal branching and growth (*Figure 4F,G*) to those with more normal ureteric bud trees. Significantly, a small number of nephrons formed ectopically away from the ureteric bud in several kidneys. The ectopic nephrons were found to be positive for Jag1, Cdh1, and laminin but lacked podocytes, as shown by the absence of Wt1 and Podxl signal (*Figure 4F,G* right panels, respectively), in accordance with our pharmacological data presented above. A few endogenous nephrons within the ureteric bud tree managed to produce proximal domains (*Figure 4F,G*). The ectopic nephrons lacking proximal domains together with the absence of β-catenin signalling in this end of the nephrons (*Figure 1*) and our inhibitor experiments, strongly suggest that increased β-catenin activity disrupted normal patterning of the developing nephron and blocked the development of the most proximal end.

## Modulating β-catenin activity shifts positional identities along the nephron without altering proliferation or apoptosis

The canonical Wnt signalling pathway is known to have a direct effect on cell proliferation (*Davidson and Niehrs, 2010*). We tested if this was part of the mechanism via which β-catenin controls nephron

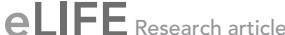

**Figure 3**. Pharmacological modulation of β-catenin signalling alters distal segment development. (**A**) *Lgr5-EGFP* expression in treated nephrons with segmentation markers. (**B**) Time-lapse analysis of treated *Lgr5*[+/EGFP-IRES-CreERT2] kidneys–arrowheads indicate developing nephrons, red-dashed line indicates ureteric bud (UB). (**C**) Mean number of *Lgr5-EGFP* positive nephrons per kidney.

*Figure 3. Continued on next page*

*Figure 3. Continued*

The following figure supplement is available for figure 3:

**Figure supplement 1**. Lgr5 expression domain in heterozygous and homozygous Lgr5[+/EGFP-IRES-CreERT2] kidneys.

patterning. Neither increasing nor decreasing β-catenin signalling levels changed the number of mitotic nuclei per nephron (*Figure 5A,B*) nor did blocking cell proliferation with methotrexate (MTX; *Chabner and Young, 1973*) affect the patterning of the nephron. Nephrons that formed in MTX were much smaller compared to control nephrons but expressed Cdh1, Wt1, and Jag1 correctly (*Figure 5C*), and the ratio of the size of the medial and proximal segment was unchanged (*Figure 5D–F*). Moreover, nephrons treated simultaneously with MTX and either CHIR or IWR produced nephrons with the same segmentation changes as caused by IWR1 or CHIR on their own (*Figure 5—figure supplement 1A*), in spite of the nephrons now being smaller as a result of the MTX treatment. Finally, MTX treatment did not affect the existence of the β-catenin activity gradient as observed with the *TCF/Lef::H2B-GFP* reporter. Moreover, co-inhibition with MTX and either IWR1 or CHIR still decreased and increased the reporter signal, respectively (*Figure 5—figure supplement 1B–D*).

Treatment with neither IWR1 nor CHIR altered apoptosis in the nephrons (*Figure 5—figure supplement 2*). Only very low levels of apoptosis were detected, primarily at the periphery of the kidneys, in line with previous data (*Foley and Bard, 2002*). CHIR treated kidneys did display some apoptotic nuclei, but these were located in the mesenchymal progenitor population surrounding ureteric bud tips. Collectively, these data indicate that changes in proliferation and apoptosis are not the driving force behind the β-catenin signalling gradient or the patterning of the nephron.

## β-catenin activity is negatively modulated in the nephron by BMP signalling which positively regulates medial segment development

In the intestine BMP negatively regulates β-catenin signalling by inhibiting the PTEN/PI3K/AKT pathway to maintain high levels of active GSK3β (*He et al., 2004*). BMP2/4/7 are expressed in the nephron (*Oxburgh et al., 2011*) and the BMP/SMAD pathway is active (*Blank et al., 2008*; *Brown et al., 2013*); albeit with an unknown function. We confirmed BMP pathway activity in the medial segment of the nephron using pSMAD1/5/8 reporter *BRE-LacZ* (*Blank et al., 2008*) (*Figure 6A*). This closely correlated with the Jag1[+] domain (*Figure 6A*). BMP activity was further confirmed using antibodies specific for pSMAD1/5/8 (*Figure 6B*).

To test if BMP signalling could inhibit β-catenin signalling via the PTEN pathway as it does in the intestine (*He et al., 2004*) we first investigated if BMP can signal via PTEN in the kidney. To do this we blocked BMP signalling at the BMP receptor (BMPR) level with inhibitor LDN-193189 (*Boergermann et al., 2010*) and PI3K with inhibitor Ly294002 (*Vlahos et al., 1994*; *Gharbi et al., 2007*). Inhibiting PI3K is equivalent to further activating any existing BMP/PTEN signalling (*He et al., 2004*). If BMP signals via PTEN, the expected outcome of BMPR inhibition would be a rise in the levels of phosphorylated PTEN (inactive) and phosphorylated AKT (active) (*He et al., 2004*), which was indeed found in nephrons and the ureteric bud (*Figure 6C*). We tested whether the inhibitors had reversible effects by releasing the kidneys from inhibition after 48 hr of treatment. A moderate recovery towards the control phenotype was observed after removal of the inhibitors (*Figure 6—figure supplement 1*). To test if BMP and PI3K signalling in the nephron affects β-catenin activity, we treated *TCF/Lef::H2B-GFP* kidneys with both inhibitors. If the system operates as in the intestine, BMPR inhibition should prevent BMP/PTEN signalling from antagonising β-catenin activity thus resulting in increased β-catenin reporter activity (*He et al., 2004*). Inhibition of PI3K should do the opposite. The total β-catenin reporter activity as measured throughout nephrons, 22 hr after nephron formation, was indistinguishable between controls and BMPR inhibited nephrons, but PI3K inhibition reduced β-catenin reporter activity levels as expected (*Figure 6—figure supplement 2A*). The time-lapse data did however visually show higher and lower *TCF/Lef::H2B-GFP* activity in the BMPR and PI3K inhibited nephrons, respectively, as we had anticipated (*Videos 8, 9* top part). Thus, we monitored the β-catenin reporter of each nephron over the whole 22 hr time-span and measured the activity along the whole distal-to-proximal axis (*Figure 6D,E*). Nephrons were randomly picked during the whole time-span of the kidney being monitored (96 hr) and the averages were determined for each treatment. This showed that the high to medium β-catenin reporter activity extended ectopically into the middle portion of the

**Figure 4**. Shifts in positional identity by altered β-catenin activity. (**A**) Model of predicted changes in segmentation if the gradient of β-catenin activity specifies positional identities in the nephron. Nephrons depicted as spheres representing renal vesicle stage. Dashed line indicates nephron segments. Gradient bar indicates β-catenin activity. (**B**) Antibody stains against segment specific markers in nephrons with different β-catenin signalling conditions. (**C**) Proximal, medial, and distal nephron domain-sizes in Control, CHIR, and IWR1 treated kidneys. Mean values and SEMs indicated within bars on graph. (**D**) qRT-PCR analysis of markers for nephron induction displayed as a heat-map with information displayed in figure. The RNA was isolated after
*Figure 4. Continued on next page*

Research article

Developmental biology and stem cells

*Figure 4. Continued*

48 hr of culture from whole kidneys. (**E**) Antibody stains on nephrons developed in isolated mesenchyme. (**F** and **G**) *Apc*[+/1638N] *Ctnnb1*[Y654/E654] kidneys where *Ctnnb1*[Y654] is the wild-type allele. Kidneys characterised using anti-Wt1, Jag1, Podxl, Lam, and Cdh1. Arrowheads and boxed area indicate ectopic nephrons in **E–F**. Dashed lines separate metanephric and mesonephric regions in **E–F**.

The following figure supplements are available for figure 4:

**Figure supplement 1**. Gradual shifts in positional identity by gentle changes in β-catenin activity.

**Figure supplement 2**. β-catenin activity dosage-dependent phenotypes in series of *Apc* and *Ctnnb1* models.

nephron when BMPR was inhibited and low levels were detected when PI3K was blocked (*Figure 6D,E* and *Video 8*, top part).

Moreover, when BMPR or PI3K were inhibited, nephrons elongated at different rates compared to controls; 11.2 µm/hr (BMPR), 10.6 µm/hr (PI3K), and 8.4 µm/hr (control) (data not shown).

Given the observed increase of the β-catenin reporter in response to blocking BMPR, by inhibiting BMPR and simultaneously activating β-catenin with CHIR, this should synergistically expand β-catenin signalling in the medial nephron segment where BMP activity was detected. The ectopic and tree-bound nephrons that formed in these dual-inhibitor conditions displayed very high β-catenin activity throughout the nephron structures (*Figure 6—figure supplement 2B* and *Videos 8, 9*, lower part). Interestingly, co-inhibition of PI3K and activation of β-catenin also positively affected β-catenin reporter activity; however, the dynamics of this was clearly different from that caused by inhibition of BMPR and activation of β-catenin with CHIR (*Videos 8, 9*, lower part). To test if segmentation was altered by inhibition of BMPR or PI3K, we stained kidneys for Wt1, Jag1, and Cdh1. Activating β-catenin and inhibiting BMPR at the same time completely removed the Wt1⁺ cells, strongly reduced the Jag1⁺ cells, and formed nephrons made of distal regions (*Figure 6—figure supplement 2C*). Activating β-catenin and inhibiting PI3K at the same time made nephrons form large medial/Jag1⁺ segments with few WT1⁺ proximal cells present (*Figure 6—figure supplement 2C*). These data suggest that PI3K inhibition positively promotes a medial fate. Inhibiting PI3K on its own strongly promoted the development of the Jag1⁺ segment and inhibiting BMPR had a slight negative effect on the Jag1⁺ segment as expected, the latter perhaps partially hidden by the overall increase in nephron length (*Figure 6F* and *Figure 6—figure supplement 2C*). qRT-PCR analyses on segment-specific genes confirmed that inhibiting PI3K had a positive effect on the medial domain; medial markers *Dll1*, *Jag1*, *Irx2*, and *Fgf8* increased, as did distal marker *Lhx1* (*Figure 6G*). Inhibition of BMPR positively affected distal markers *Bmp2* and *Lgr5* expression and also some medial markers *Heyl* and *Irx2* (*Figure 6G*). *Pax2* and *Wnt4* were both down-regulated. Co-inhibiting BMPR and PI3K resulted in nephrons highly similar to those in just PI3K-inhibitor conditions confirming that PI3K is downstream of BMPR and that PI3K can positively affect the medial segment independently of BMPR/SMAD (data not shown).

Since both β-catenin signalling and nephron patterning was modulated via the PI3K pathway, we tested whether AKT positively regulates β-catenin signalling via phosphorylation of β-catenin at Ser552, as demonstrated elsewhere (*Fang et al., 2007*). β-catenin phosphorylated at Ser552 was localised primarily to the apical surfaces of the nephrons in the distal and medial segments (*Figure 6H*). In a subset of cells, the phosphorylated Ser552 β-catenin was detected at lateral and basal surfaces. 80% of such cells were found in the distal and medial segments (45/56 positive cells counted in five nephrons) and the remaining 20% in proximal cells. Unlike phosphorylated PTEN and phosphorylated AKT, which responded to PI3K and BMPR inhibition, phosphorylated Ser552 β-catenin did not change on inhibition of either (data not shown).

These data indicate that the positional identities in the nephron can be modified through BMP mediated PI3K/AKT signalling and that simulated increased BMP signalling (via inhibition of PI3K) leads to reduced β-catenin signalling, whereas simulated decreased BMP signalling (via inhibition of BMPR) results in the opposite effect.

## Low β-catenin and PI3K activity partially rescue defective Notch-signalling

To date, the only other major pathway known to control nephron patterning is the Notch-signalling pathway (*Cheng et al., 2007*). Notch-signalling specifies the proximal and medial identity, but it can

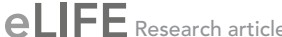

**Figure 5**. Modulating β-catenin activity shifts positional identities along the nephron without altering proliferation. (**A**) Proliferation in *TCF/Lef::H2B-GFP* expressing nephrons treated with CHIR and IWR1. Nephron axis–dashed white line. Phosphorylated Histone 3 used as a marker for mitotic cells. (**B**) Quantification of mitotic nuclei per nephron. (**C**) Effect of Methotrexate (MTX) on nephron development and patterning. Nephrons outlined with white dashed line. (**D–F**) Measurement of Jag1+ and Wt1+ segment sizes of nephrons in control, 250 nM Methotrexate (MTX), and 500 nM MTX conditions. All error bars indicate SEM. p-values generated using Student's *t* test. Scale bars and antibodies as indicated on fields. UB–ureteric bud.

*Figure 5. Continued on next page*

*Figure 5. Continued*

The following figure supplements are available for figure 5:

**Figure supplement 1**. Modulating β-catenin activity shifts positional identities along the nephron regardless of proliferation levels.

**Figure supplement 2**. Modulating β-catenin activity shifts positional identities along the nephron regardless of apoptosis levels.

also block the adjacent glomerular identity when over-activated in these cells (*Cheng et al., 2007*; *Boyle et al., 2011*). In many tissues, Notch and Wnt pathways act both in synergy and in opposition (*Andersson et al., 2011*), and Notch has also been reported to antagonise the PTEN/PI3K/AKT pathway in clear-cell renal-cell carcinoma (*Liu et al., 2013*). Our data already pointed to a negative correlation between PI3K activity and the expression of Notch ligands *Dll1* and *Jag1*. Furthermore, we showed activity of the BMP pathway in the segment affected when Notch is absent. Because we have shown that decreased β-catenin activity enhances the formation of the glomerular segment and PI3K inhibition promotes the medial segment, we tested if inhibition of β-catenin or PI3K could rescue these respective identities when Notch activity is lost. We first inhibited Notch using the γ-secretase DAPT as previously published (*Cheng et al., 2003*). This resulted in a loss of proximal and medial structures and glomerular precursors (positive for Podxl and *Lotus tetragonolobus* lectin (LTL), *Figure 7A*) and these effects mimic those phenotypes seen in *Notch2⁻ᐟ⁻* animals (*Cheng et al., 2007*). Removal of the inhibitor led to a partial recovery of proximal segments (*Figure 7—figure supplement 1*). Co-inhibition of β-catenin and Notch-signalling resulted in the reappearance of nephrons positive for LTL and Podxl that displayed glomerular structures and medial and proximal development (*Figure 7A*). The Wt1$^{+/GFP}$ reporter kidneys in time-lapse (*Figure 7B*) and antibody stains that identify these segments (*Figure 7C*) confirmed that by decreasing β-catenin activity in Notch-inhibited kidneys we partially rescued the absence of normal Notch-signalling. The inhibition of β-catenin did not restore the expression of key Notch target genes (*Jag1*, *Dll1*, *Heyl*, *Hey1*) (*Figure 7D*). Co-inhibition of PI3K and Notch also resulted in a partial rescue (*Figure 7—figure supplement 2A* and *Videos 10, 11*). Wt1$^{+}$ cells were not evident but Jag1 was again expressed, indicating that PI3K exerts an effect specific to the medial segment (*Figure 7—figure supplement 2A*). We assessed how inhibition of Notch, with or without inhibition of β-catenin and PI3K signalling, affected β-catenin activity using the β-catenin reporter *TCF/Lef::H2B-GFP*. Inhibition of Notch resulted in stunted nephrons without Wt1$^{+}$ or Jag1$^{+}$ cells with medium to high level of β-catenin activity throughout (*Figure 7—figure supplement 2B–C*; *Video 11*). Co-inhibition of Notch and β-catenin signalling strongly decreased the β-catenin activity and Wt1$^{+}$ proximal cells were again present. Co-inhibition of Notch and PI3K increased the size of the nephrons and Jag1$^{+}$ cells were again present and the β-catenin reporter was slightly decreased. Triple inhibition of Notch, PI3K, and β-catenin signalling did not result in an improved rescue (data not shown). Together, these data show that proximal and medial nephron cells require Notch signalling and simultaneously need β-catenin and PI3K signalling to be kept at low levels, respectively.

## Discussion

### An integrated model for β-catenin controlled nephron patterning

A major outstanding question in kidney development has been to understand how the nephrons are patterned. It has been shown that Wnt via canonical β-catenin signalling induces nephrons to form and through non-canonical pathways regulate the nephron mesenchymal-to-epithelial transition as well as subsequent tubule elongation (*Davies and Garrod, 1995*; *Kuure et al., 2007*; *Park et al., 2007*; *Karner et al., 2009*; *Burn et al., 2011*; *Tanigawa et al., 2011*). Our data now show that β-catenin activity is essential throughout nephron development by regulating the patterning of the nephron through interactions with the BMP, PTEN/PI3K, and Notch pathways (*Figure 8*).

Our data combined suggest a model where a gradient of β-catenin activity specifies positional identities along the nephron axis. Additional signal transduction pathways modulate local β-catenin activity levels. Its action has a differential effect on nephron segments by controlling differentiation, maturation, and their size. While cell proliferation and apoptosis are likely to be essential in other aspects of nephrogenesis, our data exclude a direct role in the initial patterning mechanism. With the

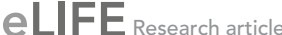

**Figure 6**. The β-catenin activity gradient is modified by changes to BMP and PI3K signalling. (**A**) BRE-LacZ pSMAD reporter shows strong labelling in medial segment; co-stained for Wt1, Jag1, Cdh1. (**B**) pSMAD1/5/8 specific antibody stain. Lines and labelling in **A**–**B** indicate different segments. (**C**) pPTEN and pAKT levels in the nephron after inhibition of BMPR with 4 μM LDN-193189 or PI3K with 20 μM Ly294002. White arrowheads and arrows indicate staining in ureteric bud and nephrons, respectively. (**D**) Time-lapse of single *TCF/Lef::H2B-GFP* positive nephrons developing from induction

*Figure 6. Continued on next page*

*Figure 6. Continued*

stage through S-shaped body stage. Nephron axis in red. Nephrons treated as specified. CM -cap mesenchyme, UB—ureteric bud. (**E**) Quantification of *TCF/Lef::H2B-GFP* intensities at different positions along the proximal–distal axis at S-shaped body stage/22 hr. Multiple nephrons used as indicated on graph. Error bars indicate SEM. Average lengths of nephrons at 22 hr for each condition are indicated on the graph. (**F**) Inhibition of BMPR or PI3K alters the medial segment negatively and positively, respectively. (**G**) qRT-PCR data of segment-specific markers and β-catenin target genes displayed as a heat-map with gene information displayed in figure. (**H**) p-β-catenin Ser552 localisation in S-shaped nephron. Antibody stains and scale bars as indicated.

The following figure supplements are available for figure 6:

**Figure supplement 1**. Rescue/reversals experiments for kidneys treated with 20 µM Ly294002 or 4 µM LDN193189.

**Figure supplement 2**. Changes to BMP and PI3K signalling alters nephron segmentation and β-catenin signalling.

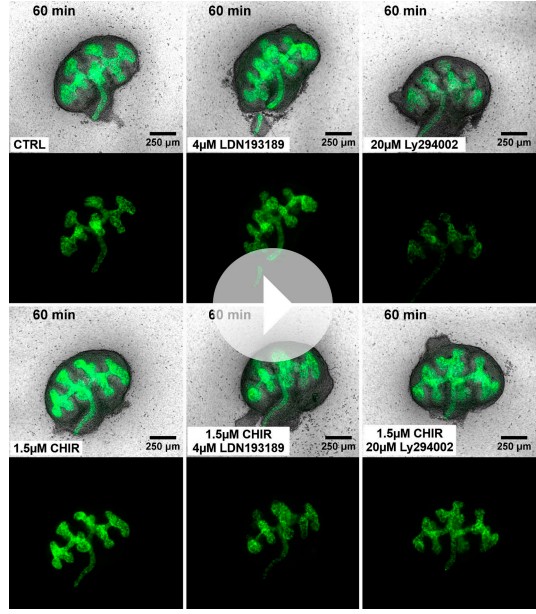

**Video 8**. Time-lapse capture *TCF/Lef::H2B-GFP* kidneys. Kidneys cultured in control medium, 4 µM LDN-193189, 20 µM Ly294003, 1.5 µM CHIR, 4 µM LDN-193189, and 1.5 µM CHIR, or 20 µM Ly294003 and 1.5 µM CHIR. Timing and scales are as specified.

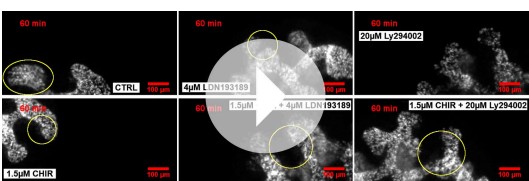

**Video 9**. Time-lapse capture *TCF/Lef::H2B-GFP* kidneys. Kidneys cultured in control medium, 4 µM LDN-193189, 20 µM Ly294003, 1.5 µM CHIR, 4 µM LDN-193189, and 1.5 µM CHIR, or 20 µM Ly294003 and 1.5 µM CHIR. Timing and scales are as specified.

current data, it is difficult to identify key downstream targets and that way describe the next level of mechanism of this process, as many targets are also widely used as segmentation markers.

## Nephrons are patterned by a gradient of β-catenin activity

Using a variety of methods, our combined data strongly suggest the existence of a gradient in β-catenin activity along the proximal–distal axis of the developing nephron. First, the *TCF/Lef::H2B-GFP* activity reporter showed a clearly recognisable visual gradient which we confirmed by quantifying the reporter signal along the nephron axis. Reporter activity is high at the distal end of the nephron, where it connects to the ureteric bud, and gradually reduces in the proximal direction, with the developing glomerulus at the extreme proximal end being devoid of β-catenin activity (*Figure 1A–C*). Second, while antibody staining for pan β-catenin showed equal expression of the protein throughout the post-MET nephron, staining with phospho-specific antibodies (indicative of protein targeted for break-down) was greatly enhanced in the proximal end of nephrons, the same segment where the reporter is least active. This also indicates that the gradient is an activity gradient, not an expression gradient. Third, the pharmacological and genetic modulation of the β-catenin signal result in a phenotype that can be predicted from the reporter and antibody data. In isolation, none of these analyses would prove the existence of a β-catenin activity gradient. Reporters of the type used here can sometimes give unreliable data, however, the model we used was found to give a very reproducible activity patterns that overlaps with many other β-catenin activity reporters (*Ferrer-Vaquer et al., 2010*). Phospho-specific antibodies are powerful tools to monitor activity of signal transduction pathways, but their use can be difficult to optimize, especially in the kidney organ culture system. Maybe surprisingly, our antibody data do not show nuclear β-catenin as would be expected

**A**

**CTRL**                                      **2µM DAPT**                          **2µM DAPT + 2µM IWR1**

*[Figure 7A: immunofluorescence images. Labels: LTL LAM CDH1; inserts labeled PODXL LTL LAM CDH1. Scale bar 250 µm; insert scale 25 µm]*

**B**

**Proximal progenitor (Wt1^GFP/+) time-lapse 0–4000 min**

| DAPT | 0 min | 1000 min | 2000 min | 3000 min | 4000 min |

| DAPT+IWR1 | 0 min | 1000 min | 2000 min | 3000 min | 4000 min |

*scale 250 µm*

**C**

**Co-inhibition of Notch and Wnt increases proximal domain development**

JAG1 WT1

CTRL

2 µM DAPT

2 µM DAPT 2 µM IWR1

*Bar chart: Mean number of Wt1+/Jag1+ nephrons*
- CTRL: 45.8, p=2.26×10⁻⁵
- 2 µM DAPT: 1.8
- 2 µM DAPT + 2 µM IWR1: 7.0, p=0.0005

**D**

**Co-inhibition of Notch and Wnt does not rescue Notch gene expression**

*Bar chart: Normalised expression (Gene/GAPDH). Legend: CTRL (dark), 2 µM DAPT (white), 2 µM DAPT +2 µM IWR1 (cyan). Genes: Jag1, Dll1, HeyL, Hey1*

**Figure 7**. Altered β-catenin activity rescues the loss of Notch. (**A**) Kidneys treated with DAPT and DAPT/IWR1 and stained for LTL, β-laminin, Cdh1, and Podxl—arrowheads indicate LTL-positive nephrons, inserts show magnified nephrons with Podxl staining for podocytes, yellow line outlines nephron tubules. (**B**) Time-lapse analysis of Wt1^+/GFP kidneys treated with DAPT and DAPT + IWR1—arrowheads show GFP^HIGH structures in developing proximal segments, red dashed line indicates ureteric bud positions (UB). (**C**) Structures positive for Jag1 and Wt1 in treated kidneys—arrowheads indicating double-positive structures. (**D**) qRT-PCR data for Notch target genes (*Jag1*, *Dll1*, *Heyl*, *Hey1*). All error bars indicate SEM.

The following figure supplements are available for figure 7:

**Figure supplement 1**. Rescue/reversals experiments for kidneys treated with 2 µM DAPT.

**Figure supplement 2**. Altered β-catenin activity or PI3K signalling rescues the loss of Notch.

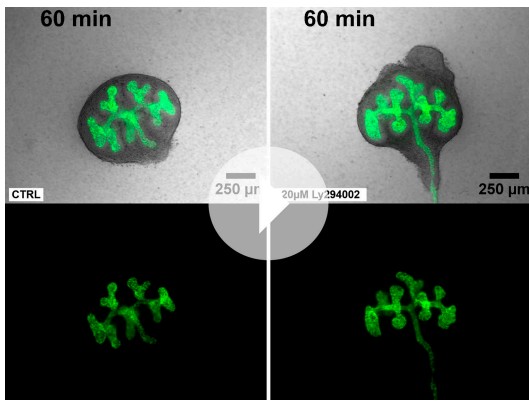

**Video 10**. Time-lapse capture of *TCF/Lef::H2B-GFP* kidneys. Kidneys in this video were cultured in control medium or 20 µM Ly294003. Timing and scales are as specified.

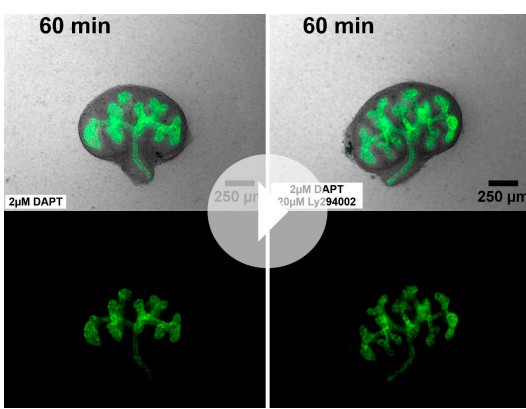

**Video 11**. Time-lapse capture of *TCF/Lef::H2B-GFP* kidneys. Kidneys in this video were cultured in 2 µM DAPT or 20 µM Ly294003 with 2 µM DAPT. Timing and scales are as specified.

for active signalling. It should be realized, however, that this is the same for the signal in the ureteric bud which is generally accepted to have active β-catenin signalling (*Bridgewater et al., 2008*; *Marose et al., 2008*). This could indicate technical limitations of the whole-mount staining, but it should also be noted that in many other developing systems it can be difficult to detect nuclear β-catenin. Even in colorectal cancer it was shown that in cells from the same tumours, all assumed to have the same initiating loss of *APC* or oncogenic activation of β-catenin, dynamic localisation changes between nucleus and cytoplasm can be found (*Fodde and Brabletz, 2007*). Similarly, expression of a constitutively activated β-catenin mutant in mice throughout the intestine results in different responses in different parts of the intestine, including different localization (varying between nuclear, cytoplasmic, or membranous; *Leedham et al., 2013*). Finally, a functional role for β-catenin in the patterning process as shown by our pharmacological and genetic data does not prove that this is dependent on an activity gradient. However, combined, these data strongly suggests that this gradient is real and functional in the patterning of the nephron.

The reporter signal showed an exponential decrease, suggesting first-order decay of a single source from the distal end. The proximal domain (Wt1+), which is normally characterized by low or absent β-catenin activity, does not form when β-catenin activity is increased, whereas it forms quicker when the activity of the pathway is decreased (*Figure 2A–D*). In the other domains of the developing nephron (Lgr5+ and Jag1+), cells also respond to the level of β-catenin activity that is forced on them (*Figure 3A,B, 4C,D*). Increasing β-catenin activity had a positive effect on the distal domain (*Figures 3B, 4C*) as expected.

Whilst the effect of decreasing β-catenin signalling was clear on the proximal segment it led to a more complex effect in the other segments. Lgr5 became expressed at earlier time-points, albeit at lower levels (both in terms of GFP fluorescence in the *Lgr5+/EGFP-IRES-CreERT2* model and also mRNA levels analysed by qRT-PCR) compared to controls and when β-catenin signalling was increased (*Figure 3B*). The tubules also elongated excessively (*Figure 4C*). We speculate that the reason for this is that decreasing β-catenin signalling might have resulted in the whole of the nephron maturing faster and therefore elongating more, however, decreasing β-catenin signalling particularly favoured the proximal domain. Since proliferation was not affected by either increasing or decreasing β-catenin signalling, the changes in tubular morphology and length must have been caused by another mechanism. The tubules appeared thinner, suggesting cellular rearrangements as a likely cause.

In addition to using genetic models we also utilised pharmacological inhibitors to enable us to finetune the β-catenin activity dosage through titration of the concentration of the compounds (*Davies, 2009*). This made it possible to inhibit as well as activate the pathway and, crucially, to achieve temporal control in the experimental system. Not only did this approach exclude the possibility that the observed phenotypes were due to cellular toxicity, but by washing away the β-catenin activator we showed that proximal nephron structures could recover and reform if the sustained β-catenin signalling, imposed on them by CHIR, was removed (*Figure 2E*). Genetic models for activation of β-catenin

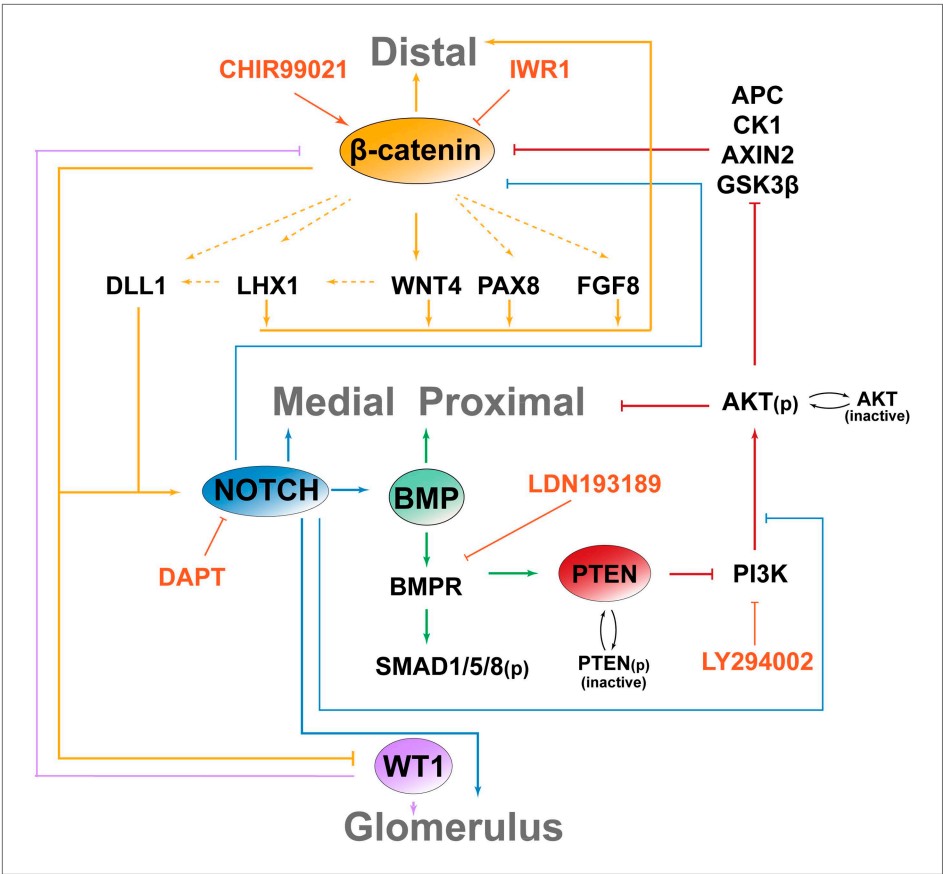

**Figure 8**. Model for molecular pathways interacting to control the patterning of the nephron. β-catenin activity is necessary to determine a distal cell identity but must be excluded from the proximal nephron. BMP/PTEN/PI3K antagonises β-catenin activity in the medial segment and positively promotes a medial fate and Notch ligand expression. Notch is essential for medial and proximal development. Glomerular progenitor cells strongly inhibit β-catenin function, possibly via a WT1-dependent mechanism as in Sertoli cells (*Chang et al., 2008*).

are available, but inhibiting the pathway is not possible with current mouse models. The series of different *Apc* and *Ctnnb1* mouse models that we employed have previously been used to show that β-catenin activity levels are selected for by different types of cancer, and these ectopic increases in signalling activity can also lead to a range of developmental defects (*Kielman et al., 2002*; *Gaspar et al., 2009*; *Buchert et al., 2010*; *Bakker et al., 2013*). Our data indicate that increasing levels of β-catenin activity through *Apc* and *Ctnnb1* mutations lead to successively more severe kidney phenotypes, but only at the highest activity–dose achieved in this study did we detect ectopic nephron development (*Figure 4F,G*). It has been known for a long time that LiCl, which is an inhibitor of GSK3β, can induce nephron formation but it also blocks MET at high concentrations (*Davies and Garrod, 1995*). This has since been confirmed by other, more specific, GSK3β inhibitors as well as genetic activation of β-catenin (*Kuure et al., 2007*; *Park et al., 2007*). Therefore, the appearance of the epithelialized, though abnormal, nephrons in the *Apc*[+/1638N] *Ctnnb1*[+/E654] mutants and the ectopic nephrons in the inhibitor studies indicate that β-catenin activation has dosage-dependent effects on the developing kidney. The absence of proximal markers in these ectopic nephrons correlated with the lack of reporter activity as well as the phosphorylation state of β-catenin in that part of the nephron. It is important to note that some nephrons within the ureteric bud trees of *Apc*[+/1638N] *Ctnnb1*[+/E654] mutant animals still managed to produce proximal domains. Whether this indicates redundancy or an ability to overcome increased β-catenin levels through ureteric bud-derived signals remains to be determined. Using cultures of spinal cord-induced metanephric mesenchymes, we also show that the observed phenotypes are not an indirect effect of changes in the ureteric bud but are a direct effect in the mesenchyme.

## Nephron patterning and the β-catenin activity gradient are independent of proliferation

Cell proliferation could be a mechanism through which β-catenin controls the patterning of the nephron. However, we found that proliferation along the nephron axis does not change in response to stimulating or inhibiting β-catenin activity, nor does blocking cell proliferation affect the patterning (*Figure 5C*), suggesting that proliferation does not play a mechanistic role in this patterning process. Even in nephrons which were treated with MTX and which were significantly smaller compared to controls did quantification of the *TCF/Lef::H2B-GFP* activity reporter show that there was a gradient in β-catenin signalling as also seen in normal nephrons. Similarly, when nephrons were treated with MTX it was still possible to induce segmentation changes by adding either IWR1 or CHIR. This implies that the underlying mechanisms behind nephron segmentation can adequately adapt to reduced cell numbers and an overall reduction in nephron size. We also found that apoptosis in nephrons was not altered by either CHIR or IWR1. Together, these findings would therefore suggest that increasing and decreasing β-catenin signalling changes segmentation by forcing cells to change their identity and that this is the most likely mechanism underlying the alterations to segmentation.

## The β-catenin gradient is spatially limited by BMP/PTEN/PI3K signalling

We also investigated how β-catenin signalling interacts with other pathways in the nephron and how the β-catenin activity gradient is limited from expanding proximally (*Figure 6*). Our data indicate that, similar to the case in intestinal stem cells (*He et al., 2004*), β-catenin activity is negatively regulated by BMP/PTEN/PI3K signalling in the medial domain of the nephron. This domain is normally positive for a BMP/pSMAD reporter and pSMAD staining. This link between BMP and β-catenin is likely to reflect a normal function of BMP in this nephron segment since inhibition of BMPR and PI3K provoked opposite effects on the medial segment and BMPR inhibition increased levels of both phosphorylated PTEN and AKT. Furthermore, inhibition of PI3K on its own made the medial segment visually grow much larger and Jag1 and other medial markers were expressed more strongly. The significance of PI3K signalling in connection with Wnt/β-catenin pathway is strongly debated (*Ng et al., 2009*), but the PI3K pathway has not only been shown to alter Wnt signalling in the gut but also in embryonic stem cells (*Paling et al., 2004*). We tested whether β-catenin was phosphorylated at Ser552 as it has been reported to be by AKT (*Fang et al., 2007*). Based on our data, we expect low levels of PI3K/AKT signalling in the medial segment, and consequently, we would therefore anticipate that β-catenin would only be phosphorylated at Ser552 at very low levels if at all. We did detect phosphorylated β-catenin at the apical surfaces and lateral surfaces mainly within the distal and medial domain but never within the nucleus (*Figure 6H*). A number of cells displayed stronger β-catenin staining which also extended to the lateral and basal surfaces. We currently do not know the significance of this and the staining pattern did not change in response to either BMPR inhibition of PI3K inhibition. It will be important to explore precisely how PI3K and BMP/pSMAD signalling regulate the Wnt/β-catenin pathway, in the nephron and elsewhere. This should be the subject of future studies.

## The medial and proximal nephron segments are dependent on Notch and low levels of PI3K and β-catenin

To date, one of the best characterised signalling pathways that have been implicated in nephron patterning is the Notch pathway (*Cheng et al., 2007*; *Boyle et al., 2011*). As Wnt and Notch signalling are tightly linked during development (*Hayward et al., 2008*), we examined if the role of Notch signalling in nephron pattern formation is linked to the β-catenin activity gradient. We found that inhibition of Notch resulted in nephrons with medium to high levels of β-catenin activity throughout. By co-inhibiting β-catenin signalling in Notch-inhibited kidneys we partially reversed the latter's patterning phenotype. This was also true of co-inhibition of PI3K and Notch. In common to inhibition of β-catenin and PI3K was that they both decreased β-catenin signalling in nephrons and positively affected the development of the proximal-most and medial segments, respectively. Inhibition of β-catenin signalling in Notch-inhibited nephrons reduced the level of β-catenin signalling and rescued the proximal-most cells whilst co-inhibition of Notch and PI3K partially rescued the medial segment. It is therefore tempting to speculate that the mechanism for these rescues is by reduction of the level of β-catenin signalling. The mechanism of the rescue needs to be further studied, particularly in light of that the expression of known Notch target genes was not rescued by the β-catenin inhibitor (*Figure 7D*).

## Implications for morphogen signalling

Wnt ligands have long been known to act as one of the classic morphogens that drive patterning during development (*Zecca et al., 1996*). To be effective as a gradient, the cells within the gradient should be able to respond to the different concentrations of extracellular Wnt ligand. In previous studies, a gradient of recombinant XWnt8 has been shown to be able to drive anterior–posterior neural patterning in dissociated *Xenopus* embryos with a corresponding gradient of β-catenin activity (*Kiecker and Niehrs, 2001*). Similarly, a gradient in nuclear β-catenin was described in the patterning of the segmentation clock that controls somitogenesis (*Aulehla et al., 2008*). Although this study analysed different β-catenin levels, in contrast to our approach, the authors focused on the extreme ends of the gradient ('off' vs 'maximal'), and it was concluded that an unidentified co-factor was essential for interpreting the Wnt gradient. We show that in the nephron variations in the levels of β-catenin activity within the observed β-catenin gradient are directly responsible for correct patterning. Whilst the phenotypic effect of the β-catenin activity gradient in patterning the nephron is obvious, the mechanistic rationale for this is unclear. It is difficult to see how the present molecular/biochemical model for β-catenin signalling would allow the existence of β-catenin dosage-dependent differential gene expression. The model predicts that once β-catenin is stabilized, it will translocate to the nucleus and activate its target genes (*Clevers and Nusse, 2012*). The data presented here, together with the 'just-right' signalling model for the role of *APC* mutations and β-catenin activity in cancer (*Albuquerque et al., 2002*; *Gaspar et al., 2009*) clearly show that β-catenin signalling is not a binary process and suggest additional levels of control of the β-catenin transcriptional output. Further, biochemical analysis of β-catenin function will be needed to elucidate this mechanism.

At present we do not know which Wnt (assuming it is a Wnt) it is that drives the gradient or how β-catenin activity is antagonised in the proximal cells. For the Wnts, none of the published *Wnt* knockout mouse models display a phenotype that would suggest a clear role in patterning of the nephron, although both *Wnt9b* and *Wnt7b*, when deleted, alter nephron morphogenesis. The *Wnt7b* knockout throughout the embryo proper or in the ureteric bud was described to lack a medulla due to disturbances of the plane of cell division, resulting in a failure of the Loop of Henle to elongate (*Yu et al., 2009*). As this phenotype was mainly analysed at later stages of kidney development, it would require the inclusion of the *TCF/Lef::H2B-GFP* reporter and analysis of appropriate markers to determine if this phenotype is linked to the patterning defects we describe here. A conditional *Wnt9b* knockout in post-MET nephrons resulted in disturbed planar cell polarity as well but resulted in expansion of tubule diameter instead of the Loop of Henle defect (*Karner et al., 2009*). Wnt9b is known to regulate planar cell polarity in the nephron, and the canonical pathway and non-canonical pathways are widely believed to be mutually competitive and inhibitory (*Grumolato et al., 2010*). Although competition is known to occur mainly at the receptor level, it could be imagined that competition for intracellular downstream targets have the consequence that when decreasing the β-catenin activity using IWR1, there is shift from the canonical pathway towards the planar cell polarity pathway resulting in the observed nephron elongation. Again, extensive further analysis of the *Wnt9b* model is required to demonstrate involvement in the processes we describe here.

We analysed *Lgr5* as a potential modifier of Wnt signalling establishing the β-catenin gradient. The nephrons forming in *Lgr5*[+/EGFP-IRES-CreERT2] homozygotes were morphologically indistinguishable to heterozygotes or control animals (*Figure 3—figure supplement 1*). This is maybe not surprising as *Lgr5* knockouts have previously been shown to be without phenotype in the intestine where it is an important marker of intestinal stem cells (*de Lau et al., 2011*).

A second question is how a single source of Wnt could establish a β-catenin signalling gradient within a morphologically convoluted tissue. It is plausible that the responsible Wnt forms an intraluminal gradient which would therefore potentially form a gradient irrespective of the actual nephron shape. It is also possible that all the nephron cells are exposed to a homogenous level of Wnt ligand and that they by cell-specific means modulate this to specific levels via, for example, Notch or BMP signalling. In the proximal domain, the β-catenin activity levels are even lower than in the medial domain, but no BMP activity is detected there. A plausible candidate for preventing β-catenin activity in this segment would be Wt1, as in in Sertoli cells Wt1 was shown to limit β-catenin activation (*Chang et al., 2008*). Whilst additional *Wnt* and other knockout studies, either conventional or conditional, might provide new clues to extend our data into a yet more complete genetic pathway, the rate of nephron formation makes this a particularly challenging process. Since patterned S-shaped body

nephrons form from mesenchymal progenitors within a 24 hr time-span, to conditionally knock out *Wnts*, it would be necessary to identify potential Cre-driving genes that are expressed at the very earliest stages of nephrogenesis but which are not active prior to nephron formation, since deletion at that time-point is likely to block the process. At present we do not know of any Cre driver that would be able to do this, so this might be a long-term goal.

## Materials and methods

### Ethics statement for experimental animals

Animals were kept at facilities at the MRC Human Genetics Unit and the University of Edinburgh (UK); Columbia University, New York (USA); Maine Medical Center, Maine (USA); the Beatson Institute, Glasgow (UK); and the Erasmus Medical Centre, Rotterdam (The Netherlands). All animal experiments and animal use were carried out according to regulations specified by the Home Office (MRC/UoE), approved by the Animal Ethics Committee and carried out in accordance with Dutch and international legislation (Erasmus) and conducted under PHS guidelines and approved by the relevant Institutional Animal Care and Use Committees (Columbia/Maine). All animal experiments were performed under Project Licenses 60/3788 and 60/4473.

### Experimental animals

Outbred CD1 animals were obtained from Charles Rivers (UK). Embryos were generated through timed mating, with noon of the day a vaginal plug was found considered E0.5. *TCF/Lef:H2B-EGFP* (Tg(TCF/Lef1-HIST1H2BB/EGFP)61Hadj) (*Ferrer-Vaquer et al., 2010*) were crossed with 129/SvEV and *Wt1*[+/GFP] (Wt1[tm1Nhsn]) (*Hosen et al., 2007*) mice were crossed with CD1. *Pax8*[+/Cre] (Pax8[tm1(cre)Mbu]) mice (*Bouchard et al., 2004*) were crossed to *Rosa26*[eYFP/eYFP] (Gt(ROSA)26[Sortm1(EYFP)Cos]) animals (*Srinivas et al., 2001*). *Six2*[+/GCiP] (*Dolt et al., 2013*) mice were crossed with Rosa26[tdRFP] (Gt(ROSA)26Sor[tm1Hjf]) (*Luche et al., 2007*). *Ctnnb1*[E654] (Ctnnb1[tm1.2Wvv]) (*van Veelen et al., 2011*), *Apc*[+/1572T] (Apc[tm2Rfo]) (*Gaspar et al., 2009*), and *Apc*[+/1638N] (Apc[tm1Rak]) (*Fodde et al., 1994*) were intercrossed as required. *Lgr5*[+/EGFP-IRES-CreERT2] (B6.129P2-Lgr5[tm1(cre/ERT2)Cle]/J) (*Barker et al., 2012*) were crossed with C57BL/6 (Harlan) or intercrossed. *BRE-Hspa1a-LacZ* were described before (*Blank et al., 2008*).

### Organ culture and pharmaceutical inhibitors

Kidneys were dissected from E12.5 embryos and cultured at 37°C with 5% $CO_2$ on 0.4 µm PET Transwell membranes (Corning, New York, NY). The culture medium consisted of MEM (SIGMA, UK M5650), 10% FCS, and 1% Pen/Strep. Isolated mesenchyme was induced with spinal cord as previously described (*Davies, 1994*; *Davies and Garrod, 1995*). Media was changed to contain IWR1 or CHIR after 48 hr and cultures kept for an additional 48 hr. Isolated Six2[+] cells were pelleted for 5 min at 800×*g* and induced with spinal cord for 24 hr. The cultures were treated with CHIR or IWR1. CHIR99021 (University of Dundee, UK), ICG001 (Enzo LifeSciences, UK), IWR1 (TOCRIS), salinomycin (SIGMA), BIO (TOCRIS, UK), Ly294002 (TOCRIS), LDN-193189 (STEMGENT, Cambridge, MA), methotrexate (SIGMA), were used as specified in the text.

### Immunofluorescent staining, image capture, and image analysis

#### Primary antibodies

anti-jagged1 (R&D Systems, Minneapolis, MN), anti-E-cadherin (BD Transduction Laboratories, UK), anti-wt1 (Santa Cruz, Dallas, TX), anti-LEF1 (Cell Signaling, The Netherlands), anti-Pax8 (Proteintech Europe, UK), anti-Pax2 (Covance, Princeton, NJ), anti-phospho-β-cat (S33, S37, T41) (Cell Signaling), anti-phospho-β-cat (S45) (Cell Signaling), pan β-cat (SIGMA), anti-dephospho-β-cat (AG Scientific, San Diego, CA), anti-podocalyxin, anti-laminin (SIGMA), LTL (Vector Labs, Burlingame, CA), anti-ZO1 (DSHB, Iowa City, IA), anti-pSMAD (Cell Signaling), anti-pAKT (Cell Signaling), 6-CF (SIGMA), PNA (Vector Labs), Annexin V (Biovision, San Francisco, CA), TUNEL (ROCHE, Switzerland), anti-phospho-β-cat (S552) (Cell Signaling).

#### Fluorescent secondary antibodies

Fluorescent secondary antibodies were purchased from Invitrogen Molecular Probes: anti-mouse IgG 488, anti-rabbit IgG 594, anti-goat IgG 488, anti-goat IgG 594, anti-mouse IgG 647, anti-rabbit IgG 488, anti-goat IgG 350.

## Immunfluorescent staining

Organ cultures were fixed in −20°C methanol or 4% PFA in 1× PBS. 4% PFA was used for all tissues with fluorescently labelled proteins. Fixed tissues were washed thoroughly in 1× PBS prior to addition of primary antibodies and incubation at 4°C O/N. Samples were again washed thoroughly in 1× PBS before incubation with secondary antibodies at 4°C O/N. Excess secondary antibodies were washed off thoroughly with 1× PBS before mounting in Vectashield (Vectorlabs).

## Microscopy and image processing

Live imaging microscopy was performed on a Nikon TiE (Perfect Focus System) with NIS-Elements 4.0. Imaging was performed using a 4× objective and a CoolSnap HQ2 CCD camera (Photometrics, Tucson, AZ). Confocal microscopy was carried out on a Nikon A1R and an N-STORM/A1 Confocal and 10×–60× objectives were used and image stitching was carried out at 10× to capture whole kidneys. Nikon NIS-Elements 4.0, Fiji (http://fiji.sc/), Volocity (PerkinElmer, Waltham, MA), and ImageJ (http://rsb.info.nih.gov/ij/) were used for image analysis and presentation. 3D videos were made in Fiji. Time-lapse videos were presented using ImageJ, Fiji, and IP-Lab. Brightness and contrast levels were adjusted in Adobe Photoshop CS5 and IP-Lab after quantification and for presentational purposes only.

## Quantification of H2B-GFP signal along the nephron axis

Kidneys were fixed to preserve H2B-GFP fluorescence using 4% PFA in 1× PBS. Antibody stains were performed as described above. Whole nephrons were scanned at 40× at 0.5 µm z-intervals. Nephrons were visualised on NIS-Elements 4.0. Using the line intensity-profile tool, 10 µm wide lines were constructed centrally through each nephron segment and the maximum pixel value was determined along the line at each z-plane. The mean of the maximum values was calculated for 10 µm intervals along the whole nephron axis. The percentage of the mean maximum value was plotted against the corresponding position along the nephron axis. A total of 11 nephrons were analysed that encompassed: renal vesicle, comma-shaped body, S-shaped body, and capillary-loop stage nephrons. For the multi-nephron analysis, the mean value of each segment (distal, medial, proximal) was calculated and combined for all the nephrons analysed. Two-tailed Student's $t$ tests were used to compare the medial with distal and medial with proximal segment.

## Quantification of phospho-β-cat (S33, S37, T41) signal along the nephron axis

Kidneys were stained for phospho-β-cat (S33/S37/T41), Jag1, and Cdh1. Scans were made every 1 µm with a 63× objective through whole nephrons with an N-STORM/A1 confocal microscope. 3D volume reconstructions were generated in IMARIS. Very bright pixels from background noise were made into a volume and masked onto the original channel and removed. Volume measurements were made in 75 × 75 × 10 µm cubes. Three cubes were fitted per nephron segment, thus nine measurements were made per nephron. The Image intensity sum was calculated and the average found per segment and across five different nephrons. Two-tailed Student's $t$ tests were used to compare the medial with distal and medial with proximal segment.

## Measuring image intensities in live nephrons

Whole E12.5 kidneys were cultured in control, LDN-193189, or Ly294002 containing media. Time-lapse imaging was performed to capture GFP and brightfield. The GFP signal was captured far below saturation levels (saturated pixels in figures and videos are present due to brightness and contrast changes performed for publication purposes) as a 16-bit image every 20 min. To measure the GFP signal intensity, individual nephrons were selected from their point of formation and monitored for 22 hr. The point of formation was determined based on the brightfield view and the GFP signal, which was detectable at low levels in the pre-epithelial cells. The earliest signs of nephron formation were when cells aggregated prior to epithelialisation (pretubular aggregates). That was chosen as 'time 0'. The 22 hr time-frame was chosen as this was the time taken for nephrons to develop to an S-shaped body stage in controls. To measure the intensity of nuclei along the nephron distal-to-proximal axis a 5-µm wide line was drawn down the length of the axis. The mean value across the line was measured along the whole length of the nephron. This gave readouts of the intensities of nuclei along the nephron and also provided the nephron lengths. Several nuclei were thus picked up within each nephron segment. The nephron axis was visible in the GFP and the brightfield channels and both were

used to accurately reconstruct the nephron axis. The intensities were plotted against the axis length in 8-bit values vs µm position. Several nephrons from separate kidneys were measured in this fashion for each condition: 11 control nephrons, 12 LDN-193189 nephrons, 11 Ly294002 nephrons. The nephron lengths were used to calculate the changes in nephron lengths between each time-point and over the total 22 hr time-period. This allowed us to calculate the rate of nephron growth. To measure the 'total intensity' at the 22 hr time-point, the intensity of all pixels within the whole nephron at that time-point was measured. For this, 17 control nephrons, 18 LDN-193189 nephrons, and 18 Ly294002 nephrons were measured. The pixel values were plotted against their percentage frequency to obtain curves displaying the frequency of GFP signal intensities.

## Identification of CHIR99021 concentrations stimulating epithelialisation and nephron induction

Kidneys were treated with a range of CHIR concentrations (1.5 µM, 3 µM, or 6 µM shown) and immunostained for Lef1, Pax2, Pax8, Chd1. Using Fiji the mean secondary antibody signal-intensity was measured within 50-µm diameter circular sections in all ectopic nephrons using two kidneys per treatment. All comparisons between treatments were made using samples that had been simultaneously stained and imaged.

## Quantification of TCF/Lef::H2B-GFP fluorescence to determine inhibitor effects

H2B-GFP fluorescence was measured in all ureteric bud tips of four kidneys per treatment. Images for measurements were captured using the same settings throughout. Statistical analysis was performed using single factor ANOVAs (α 0.05) for single time-points (0 hr, 24 hr, 48 hr).

## Glomerular maturation rates

Kidneys were cultured and imaged as described above. Kidneys were scored for the development of crescent-shaped glomeruli and the time was recorded. Statistical analysis of timing of crescent-shaped glomerular formation was carried out using two-tailed Student's $t$ tests of each group compared to controls. 5 $Wt1^{+/GFP}$ (Wt1$^{tm1Nhsn}$) kidneys were tested per group.

## Analysis of segment transformations and sizes

Confocal scans were carried out on kidneys cultured in specified conditions. Wt1, Jag1, Cdh1 immunofluorescent stains were used to measure the size of each nephron domain. Fiji software was used to measure the length of each domain. Where overlap of markers was seen, the mid-point of overlaps was taken as end of each domain. The mean domain-size was calculated and separate two-tailed Student's $t$ tests were used to analyse statistical significance compared to controls.

## Quantification of H3(S10$_P$) nuclei

Confocal scans were performed on *TCF/Lef:H2B-EGFP* kidneys cultured in the specified conditions. Kidneys were stained for H3(S10$_P$), Cdh1, and Hoechst. Positive nuclei were counted within nephrons. 21–22 nephrons were counted per condition.

## TaqMan qRT-PCR

For TaqMan experiments, samples were placed in RNALater (Ambion). A minimum of three kidneys were used per replicate of each condition and all experiments were performed in triplicates. RNA was isolated from cultured kidneys using RNAeasy micro kits (QIAGEN, The Netherlands). cDNA was made using SuperScript II (Invitrogen, Carlsbad, CA) and random primers (Promega, Madison, WI). Assays were done with the Universal Probe Library (Roche) and were designed using the Universal Probe Library Assay Design Center (http://www.roche-applied-science.com/sis/rtpcr/upl/index.jsp?id=UP030000); primer and probe details can be found in *Supplementary file 1*. All TaqMan assays used a mouse GAPDH internal control (Roche). Heat-maps were generated using the matrix2png interface (http://www.chibi.ubc.ca/matrix2png/bin/matrix2png.cgi).

## Abbreviations

GSK-3β, glycogen synthase kinase 3 beta; WNT, wingless-related MMTV integration site; Cited1, Cbp/p300-interacting transactivator with Glu/Asp-rich carboxy-terminal domain 1; Wt1, Wilms' tumour 1 homologue; Six2, sine oculis-related homeobox 2; Jag1, jagged 1; Ctnnb1, catenin (cadherin associated

protein) beta 1 (β-catenin); PTEN, phosphatase and tensin homolog; PI3K, phosphatidylinositide 3-kinase; AKT, thymoma viral proto-oncogene 1; BMP, bone morphogenetic protein; Apc, adenomatosis polyposis coli; Lgr5, leucine rich repeat containing G protein coupled receptor 5; Lhx1, LIM homeobox protein 1; Fgf8, fibroblast growth factor 8; Pax8, paired box gene 8; UB, ureteric bud; Pod., podocyte; Pt., parietal epithelium.

## Acknowledgements

We thank the members of the Hohenstein lab and Nick Hastie for valuable discussions. We thank Shahida Sheraz, Anna Thornburn, and Rachel Berry for help with animals, Paul Perry, Matthew Pearson, and Ann Wheeler for help with confocal and time-lapse imaging, Kat Norrby for providing reagents, and Elizabeth Freyer for FACS sorting. NOL was supported by the National Centre for Replacement, Refinement, and Reduction of Animals in Research (Grant 94808). The Roslin Institute receives strategic funding from the BBSRC.

## Additional information

### Funding

| Funder | Grant reference number | Author |
|---|---|---|
| National Centre for the Replacement, Refinement and Reduction of Animals in Research | 94808 | Nils O Lindström |
| Biotechnology and Biological Sciences Research Council | BB/J003416/1 | Nils O Lindström, Jeanette A Johansson, Denis J Headon, Peter Hohenstein |

The funders had no role in study design, data collection and interpretation, or the decision to submit the work for publication.

### Author contributions

NOL, Conception and design, Acquisition of data, Analysis and interpretation of data, Drafting or revising the article; MLL, C-HC, Performed the Annexin V assays; SFB, Provided *TCF/Lef:H2B-EGFP* animals; JAJ, DJH, Helped with the analysis of BMP signalling; ERMB, RS, Provided *Apc* and *Ctnnb1* transgenic animals; RAR, OJS, Provided *Lgr5*[+/EGFP-IRES-CreERT2] animals; MJK, LO, Provided BRE-LacZ animals; JAD, Conception and design, Analysis and interpretation of data; PH, Conception and design, Analysis and interpretation of data, Drafting or revising the article

### Author ORCIDs

Peter Hohenstein, http://orcid.org/0000-0001-8548-4734

### Ethics

Animal experimentation: Animals were kept at facilities at the MRC Human Genetics Unit, The Roslin Institute and the University of Edinburgh (UK), Columbia University, New York (USA), Maine Medical Center, Maine (USA), the Beatson Institute, Glasgow (UK), and the Erasmus Medical Centre, Rotterdam (The Netherlands). All animal experiments and animal use was carried out according to regulations specified by the Home Office (MRC/UoE), approved by the Animal Ethics Committee and carried out in accordance with Dutch and international legislation (Erasmus) and conducted under PHS guidelines and approved by the relevant Institutional Animal Care and Use Committees (Columbia/Maine). All animal experiments were performed under Project Licenses 60/3788 and 60/4473.

## Additional files

### Supplementary file

• Supplementary file 1. Primers and UPL probes are used in qRT-PCR analysis.

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
