## [Decision Letter]

Thank you for sending your work entitled “Integrated β-catenin, BMP, PTEN, and Notch signalling patterns the nephron” for consideration at *eLife*. Your article has been favorably evaluated by Janet Rossant (Senior editor), a Reviewing editor, and 4 reviewers.

The Reviewing editor and the reviewers discussed their comments before we reached this decision, and the Reviewing editor has assembled the following comments to help you prepare a revised submission.

All four reviewers found your work to be comprehensive in scope and for the most part found the data to justify the conclusions drawn, and the manuscript well-written. In fact, one of the strengths cited by multiple reviewers was the comprehensiveness of the study relative to other existing work on the signaling pathways controlling kidney morphogenesis. That said, although your work is in principle suitable for publication in *eLife*, there are a few issues raised by the reviewers that merit your attention prior to further consideration.

Below, I delineate the issues raised that if satisfactorily addressed would allow us to move forward. In the reviewers' view, these should not be insurmountable hurdles for you, and we'd expect that you should be able to address the issues expeditiously:

Experimental issues:

Nephron patterning is assessed using several parameters: maturation, segment length, protein expression and gene expression. The results of these distinct assessments are not explicitly compared and integrated into one conclusion.

A changing set of segment specific markers is used for nephron segment identification (e.g. Figures 4 and 6).

Although some of the pathway modulations are performed intermittently (canonical Wnt activation by CHIR, followed by CHIR withdrawal, Figure 2), other pathway modulations are performed only continuously (e.g. BMP).

Reviewer 2 points out that β-catenin staining in Figure 1 and Figure 1–figure supplement 2B, D only shows membrane bound β-catenin and not the nuclear β-catenin necessary for the transcription of Wnt target genes. In addition, the presence of phosphorylated β-catenin in the developing nephron in Figure F appears to be limited to one cell, which is insufficient to support the claim of a gradient of β-catenin phosphorylation (422). In addition, the supposed gradient of phosphorylated β-catenin along the tubule is insufficiently demonstrated by Figure 1–figure supplement 2B. These figures do not sufficiently explain the β-catenin activity gradient. Reviewer 1 points out an issue with the reliability of WNT/beta catenin reporters in detecting WNT signaling and that they should not be considered definitive (Figure 1). Moreover, the images don't completely agree with the quantifications shown. On this point, multiple reviewers have raised questions as to how a single WNT source might generate such a gradient and whether a gradient has been definitively shown. A final issue relating to this point is what (where) is the source of the Wnt?

Continuing along this line of ambiguity is the issue of why does both decrease and increase of WNT give increased maturation rate? (Figure 2). And why is the CTRL showing no crescent-shaped glomeruli until 3500 minutes (Figure 2)?

Several of your reviewers also felt that your section on the link to Notch signalling is somewhat confusing and is not well-integrated with the first two major findings. This section needs bolstering with additional data to make a more compelling link.

Reviewer 3 points out that it would be straightforward to relate to other systems your finding that the PI3K pathway regulates beta-catenin. One way to explore these parallels is to evaluate whether Akt activates beta-catenin by phosphorylation at Ser552 in the nephron.

Writing/Text issues:

The number and organization of the figures, movies and supplementary figures interferes with the readability of the manuscript. Reducing the number of main and supplementary movies, figures and panels will enhance the legibility of the manuscript.

The presentation of the data and conclusions drawn needs to be more structured, and some of the statements, e.g. those of reviewer 2, need to be more explicitly stated upfront (issues with interpreting the Lgr5 data, fact that Wnt KO's have no consequences thus far to nephron development, etc.). Reviewers 1-3 provided some specific recommendations for recrafting the text to more coherently flow for the reader.

Please pay attention to reviewer 3's comment that more is known regarding nephron development than the authors sometimes claim, as this is important to correct. Among them, the example cited is a good one, point out your claim that “The nephron β-catenin activity gradient is to our knowledge the first instance of a β-catenin activity gradient being described in any system”, when in fact this has been shown for multiple systems, including the hair follicle, hematopoietic and intestinal stem cell lineages.

The review comments below should put this summary into context:

Reviewer #1:

This is a very interesting manuscript that defines an important role for canonical WNT signaling in the patterning of the nephron. Some evidence is provided suggesting that there is normally a WNT gradient going from distal (strong) to proximal (weak). Increasing WNT, chemically or genetically, results in increased distal and decreased proximal regions of the nephron. Decreasing WNT gives more complicated results, although it is argued that there is a more rapid maturation of the proximal nephron.

The work is quite thorough in many respects, for example in using a multitude of chemical modifiers of WNT signaling, as well as genetic approaches. The characterization of resulting phenotypes is also generally excellent, with many outstanding images provided and a reasonable set of markers employed.

There are some significant unanswered questions that are raised by the results and the proposed model. As mentioned in the paper, which WNT is responsible for the proposed gradient? And why don't any Wnt knockouts show the predicted nephron patterning defect? Further, why does knockdown of WNT in the proximal region of the nephron, where WNT is already proposed to be extremely low, partially rescue the Notch mutant loss of proximal structures?

Despite these questions there is a wealth of interesting data presented. There is a real lack of understanding of the forces that pattern the nephron and this paper begins to fill that important gap.

*Reviewer #2*:

The authors have investigated the role of canonical Wnt signaling in nephron patterning during murine renal development in ex vivo culture experiments. In addition, the effect of BMP and PTEN/PI3K/AKT signaling on nephron patterning as well as the interaction between these pathways was investigated.

Extensive intervention studies have been undertaken with both pharmacological compounds as well as mutant mouse models, and the effect on patterning is assessed morphologically, immunohistochemically and by gene expression analysis. The results provide relevant and new insight into mechanisms involved in nephron patterning.

However, I do have serious concerns regarding the way the data is presented and some concerns about the coherence of the experiments.

Major comments:

The number and organization of the figures and movies interferes with the readability of the manuscript. Reducing the number of movies, figures and panels will enhance the legibility of the manuscript.

The presentation of the data and conclusion should be more structured:

Conclusions are presented inconsistently in the manuscript body, the figure legends (Figure 2—figure supplement 1) and the figure itself (Figure 6—figure supplement 1).

Paragraphs in the Results section sometimes end with a conclusion and sometimes with just research results. The manuscript contains redundant information and conclusions: e.g.: “Since proliferation was not affected by either increasing or decreasing β-catenin signaling” and “we found that proliferation along the nephron axis does not change in response to stimulating or inhibiting β-catenin”, both in the Discussion, and “phospho- and dephospho-specific antibodies (indicative of protein targeted for break-down and active 400 β-catenin respectively)” and “antibodies for β-catenin that was already tagged for degradation”, in the Discussion and in the Results respectively. Improving the cohesion (in the presentation) of the different experiments would strengthen the manuscript.

Nephron patterning is assessed using several parameters: maturation, segment length, protein expression and gene expression. The results of these distinct assessments are not explicitly compared and integrated into one conclusion.

A changing set of segment specific markers is used for nephron segment identification (e.g. Figures 4 and 6).

Although some of the pathway modulations are performed intermittently (canonical Wnt activation by CHIR, followed by CHIR withdrawal, Figure 2), other pathway modulations are performed only continuously (e.g. BMP). Vague remarks should be replaced by more clearly defined statements.

Different terms are used interchangeably in the Discussion section, but remain devoid of proper definition. Please clarify how these terms relate to the parameters measured (see above): “more efficient formation of the proximal domain”, “a strong positive effect on the development of the medial segment”, “increased rate of maturation”, “change in cell identity”.

In the Discussion section, the sentence “Exploring the roles… of future studies”: In the Results, the authors mention that Lgr-5 is a β-catenin target, however Lgr5 is also a Wnt receptor component that can enhance canonical Wnt signaling (28). The Lgr-5 expression (gradient) might therefore not be the result but the cause of the β-catenin gradient. The latter hypothesis is rejected by the experiments with Lgr-5 KO kidneys. This consideration should be mentioned more explicitly in the manuscript.

The authors state that Figure 1 suggests that β-catenin activity is induced by (factors secreted by (?)) a single source with a first order decay. This concept is difficult to visualize due to the spiral morphology of the immature nephron. Please clarify. (One might however hypothesize that the β-catenin activity-inducing signal is propagated intraluminally.)

β-catenin staining in Figure 1 and Figure 1–figure supplement 2B, D only shows membrane bound β-catenin and not the nuclear β-catenin necessary for the transcription of Wnt target genes. In addition, the presence of phosphorylated β-catenin in the developing nephron in Figure F appears to be limited to one cell, which is insufficient to support the claim of a gradient of β-catenin phosphorylation made in the Discussion section. In addition, the supposed gradient of phosphorylated β-catenin along the tubule is insufficiently demonstrated by Figure 1–figure supplement 2B. These figures do not sufficiently explain the β-catenin activity gradient.

*Reviewer #3*:

In this manuscript, the authors investigate pattern formation of the nephron. Although nephron development is a relatively neglected area of research, more is known regarding its development than the authors sometimes claim. Nonetheless, this manuscript provides a far more detailed and integrated view of nephron pattern formation than any other study to date. First, the authors conduct a detailed series of experiments showing that a beta-catenin activity gradient determines positional identity in a morphogenetic manner. This is shown primarily by using pharmacological inhibitors/activators of Wnt signaling along with a reporter of beta-catenin activity. The authors do a nice job verifying the specificity and fidelity of their system. Importantly, this pharmacological study is further supported genetically using various hypomorphic alleles of beta-catenin. The authors then show that, similar to the intestine, nephron patterning is further regulated through a BMP/PI3K network, which acts in modulating the more central-acting beta-catenin activity. Given the structural and functional similarities between the nephron and intestine, this is an attractive model. The authors also attempt to link their data with Notch signaling, which is also known to regulate nephron patterning. This section is somewhat confusing and is not well-integrated with the first two major findings. Overall, the authors have provided a more comprehensive study of nephron patterning than any previous study, and their study sheds some light on its subject that could not be gleaned from the current literature. However, some questions/concerns should be addressed prior to publication.

Minor comments:

The results showing no changes in proliferation or apoptosis should be better explained and integrated into the manuscript. Do the authors suggest this implies that differentiation is mainly responsible for the patterning effects? Actually, changes in proliferation and apoptosis are well-known mechanisms for regulating morphogenesis (along with differentiation). At any rate, the authors should better clarify the purpose of these experiments and what they conclude from the results.

The finding that the PI3K pathway is regulating beta-catenin is interesting. These pathways are known to interact, especially in the intestinal and hematopoietic systems, where Akt can activate beta-catenin by phosphorylation at Ser552. It would be interesting to see whether and where pS552-bcat positive cells were present during nephron patterning. Such data could further support the author's model.

The authors sometimes discount previous findings. For instance, they claim that “The nephron β-catenin activity gradient is to our knowledge the first instance of a β-catenin activity gradient being described in any system”. This is surprising since such gradients have been described in multiple systems, including mammals.

The authors should consider either better integrating and explaining the Notch data or removing this section from the manuscript. It would be appropriate to discuss what's known about Notch and how it might relate to their new findings in the Discussion section; however, the current presentation tends to leave the reader more confused than enlightened.

---

## [Author Response]

*Experimental issues*:

*Nephron patterning is assessed using several parameters: maturation, segment length, protein expression and gene expression. The results of these distinct assessments are not explicitly compared and integrated into one conclusion*.

We have added a combining/concluding paragraph to the beginning of the Discussion. We have limited this to what are in our opinion the clearly proven facts. Other parts we consider more speculative are discussed elsewhere in the Discussion.

*A changing set of segment specific markers is used for nephron segment identification (e.g.*
Figures 4 and 6*)*.

In the original manuscript we chose markers that best illustrated the phenotype described in the specific part of the manuscript. In particular, in cases where markers can be used in different ways, for instance genes that are known to be downstream targets of the pathways, we study as well as markers for spatial segments of the nephron, we thought this would help to illustrate our points. However, we appreciate that this has led to confusion for the reader. We now provide a consistent set of markers in the different experiments.

In the original manuscript, we presented these data as overlapping line diagrams in different colours. However, the expression patterns of the additional genes we now show made these very confusing figures, even if different colours were combined with different line styles. Eventually we decided to present these data in the revised manuscripts in the form of heat maps. Although these have the disadvantage of that we cannot include statistical information in the form of error bars, this presentation is the clearest in showing the biological message coming from the experiments.

*Although some of the pathway modulations are performed intermittently (canonical Wnt activation by CHIR, followed by CHIR withdrawal,*
Figure 2*), other pathway modulations are performed only continuously (e.g. BMP)*.

We now provide the reversal experiments for the important inhibitors we use in the manuscript. In all but one of the inhibitors (the exception being IWR1) this resulted in at least partial reversal of the phenotype. Please note that a failure to reverse the phenotype does not necessarily imply the experiment is flawed. For instance, in some case inhibitors can bind irreversibly to their target making phenotypic rescue impossible. In other instances, the change that is made might be of such a nature that the patterning would simply not be able to reverse. IWR1, for example, causes the nephrons to mature quicker. Once segments have fully matured, they might no longer have the capacity to generate those segments that didn’t form due to the inhibitor’s effect. Therefore, withdrawal of the inhibitor would simply not allow for a reversal of the effects.

*Reviewer 2 points out that* β-*catenin staining in*
Figure 1
*and Figure 1–figure supplement 2B, D only shows membrane bound* β-*catenin and not the nuclear* β-*catenin necessary for the transcription of Wnt target genes. In addition, the presence of phosphorylated* β-*catenin in the developing nephron in Figure F appears to be limited to one cell*, *which is insufficient to support the claim of a gradient of* β-*catenin phosphorylation. In addition, the supposed gradient of phosphorylated B catenin along the tubule is insufficiently demonstrated by Figure 1–figure supplement 2B. These figures do not sufficiently explain the* β-*catenin activity gradient.*

We appreciate the original data using the phospho- and dephospho-specific antibodies on its own was insufficient to prove the existence of a β-catenin activity gradient, and for this reason we interpreted it in the context of all other data (reporter and pathway modulation). Unfortunately, these antibodies are very difficult to use in the whole mount kidney organ culture system (we are not aware of other publications that show their use in this) and optimization options are limited. We did succeed in improving the data with the Ser33/37/Thr41 phospho-specific antibody and these data are now provided. To further support the activity gradient this signal is now quantified in different parts of the nephron which confirms the different signals in different parts of the nephron.

We did not succeed in further improving the pSer45 and dephospho-specific data. As we do not want to present data that has, for technical reasons, limited relevance these have now been removed from the manuscript to reduce the complexity of the presented data. We stress the pSer33/37/Thr41 data should still be considered in the context of the complete manuscript, not as stand-alone data. This is now more clearly discussed in the Discussion section.

As for the lack of nuclear β-catenin, although we cannot exclude technical complications with the whole organ staining, the model that active signalling automatically leads to and requires nuclear localization is likely an oversimplification. For instance, in colorectal cancer (the archetypal example of a tumour caused by activation of this pathway through either loss of APC or activating mutation in β-catenin) it has been shown that within the tumour there is a dynamic change in localization in different parts of the same tumour although all cells will carry the same cancer-inducing mutation (PMID 17306971). Likewise, in mice expressing the same activating *Ctnnb1*^ex 3^ mutation throughout the intestine, the biological effect *as well as the intracellular localization of the mutant β-catenin* varies in different parts of the gut (PMID 22287596). This is now also discussed in the manuscript. Therefore, although we agree this is an important observation, in our opinion it does not necessarily disprove the existence of the gradient.

*Reviewer 1 points out an issue with the reliability of WNT/beta catenin reporters in detecting WNT signaling and that they should not be considered definitive (*Figure 1*)*.

We fully agree with this reviewer that activity reporters need to be used with common sense. Whereas we have chosen to use an artificial mini-construct reporter, other labs prefer to use the *Axin2-lacZ* knock-in line (*Axin2* being an endogenous β-catenin target). Our preference for the mini-construct is based on the following: i) the relevance of endogenous target genes can differ in a cell type-dependent manner. For instance, c-Myc is a key β-catenin target in colorectal cancer (PMID 17377531) but Ccnd1 is not (PMID 15946945). If endogenous transcriptional responses are not uniform, reporters based on them will not be reliable in their responses; ii) endogenous reporters will only be fully reliable if β-catenin is the only pathway that controls its expression. This is unlikely to be the case. A reporter based on an endogenous β-catenin target gene can in that case truthfully report increased activity of the pathway, but reduction is not necessarily detected as other pathways could maintain its expression. In this respect, the simplicity of mini constructs as we used is a clear advantage, and we show that IWR1 treatment results in the expected decrease in signal intensity; iii) the use of *lacZ* in the *Axin2* model gives an additional layer of arbitrary experimental conditions. Staining time can be chosen at will and the *lacZ* signal rapidly reaches a maximum. For subtle differences, as we show here, a fluorescent reporter and quantification methods as we employ them here is far superior.

We do emphasize that none of the experiments on the gradient should be seen in isolation (see the previous point), and this is now discussed in more detail.

*Moreover, the images don't completely agree with the quantifications shown*.

The detectors of the imaging systems, and therefore quantification of the signal, have a far better dynamic range than computer screens and printers. RGB images have a value range 0-255, with 255 being a fully saturated pixel. In the example of, for instance, the 39x difference between the proximal and distal end of the nephron (in the Results section), this would mean that an almost-but-just-not saturated pixel at the high end (value 254) would give a pixel with value 6.5 at the low end. On every printer and all but the best calibrated screens this would simply be black. For this reason we have chosen to quantify as much of our data as possible, it is the only way to see the nuances in the actual data. Visual recognition is due to the technical limitations of screens and paper simply not good enough. The fact that many structures are in close proximity of the ureteric bud (with very high reporter activity) makes it even more difficult to present proper imaging data that shows all the nuances that exist in the data. However, because this was an issue raised by several reviewers we chose to include additional data to more clearly display the different intensities. Figure 1—figure supplement 1 now shows the S-Shaped body nephron in Figure 1 but at four differently enhanced brightness levels. In the brightest field, we now indicate that the podocytes are very weakly positive whilst other areas of the kidney are negative indicating that this is a real difference.

*On this point, multiple reviewers have raised questions as to how a single WNT source might generate such a gradient and whether a gradient has been definitively shown*.

The details of the mode of Wnt secretion and spreading are hotly debated in the literature. We can only speculate that it is unlikely the (hypothetical) Wnt molecule would be traveling on the outside of the nephron, as this would imply movement through the basement membrane in the responding cell. This would indeed also be complicated, as implicated by the reviewers, that the convoluted 3D structure of the nephron would be difficult to imagine in a gradient, especially in the context of the growing organ. Intraluminal transport however would, by definition, follow the shape of the nephron irrespective of the shape and organ as a whole. It would allow the Wnt signal to be taken up by receiving cell via the apical side (i.e. no basement membrane), and would fit with the observation that cilia are present on that side of the cell. We could have included this speculation in the Discussion, but as at the moment our manuscript is written from the β-catenin angle and not the Wnt angle we have not yet done so. The second issue on the gradient is discussed above.

A final issue relating to this point is what (where) is the source of the Wnt?

We have extended the Discussion with some speculation of a potential role of *Wnt7b* and *Wnt9b* in the patterning of the nephron as we describe, and indicate that extensive additional analyses beyond the scope of this work are required to prove or disprove this. We also like to stress that because of this we refer consistently to a β-catenin activity gradient, not a Wnt gradient. We even mention that at present we cannot exclude the possibility this is a hypothetical Wnt-independent β-catenin function (though we think this is highly unlikely).

*Continuing along this line of ambiguity is the issue of why does both decrease and increase of WNT give increased maturation rate? (*Figure 2*)*. *And why is the CTRL showing no crescent-shaped glomeruli until 3500 minutes (*Figure 2*)?*

We think there is a misunderstanding here. Only the decreased β-catenin signalling results in increased maturation rate (Figure 2). We agree that the labelling on the figure and perhaps also the terminology used was confusing. In the controls, there are nephrons continuously forming during the experiment. However, only those that appear mature in their morphology are counted. In IWR1 treated kidneys the glomeruli reached this mature stage faster compared to controls. The timing in this experiment is consistent with that in other experiments we show, e.g. Figure 2. There we show kidneys after 2880 min (48 hrs) of culture. Whilst there are several Podxl positive structures in the control, Figure 2 illustrates that these are not yet at the stage of 'crescent-shaped' glomeruli. We have now attempted to make the figure more accessible and the figure legend clearer.

*Several of your reviewers also felt that your section on the link to Notch signalling is somewhat confusing and is not well-integrated with the first two major findings. This section needs bolstering with additional data to make a more compelling link*.

We thank the reviewers for this criticism. We have added data on the β-catenin activity reporter under Notch/β-catenin and Notch/PI3K modulated conditions. These data have made the link between Notch and the other pathways considerably stronger.

*Reviewer 3 points out that it would be straightforward to relate to other systems your finding that the PI3K pathway regulates beta-catenin. One way to explore these parallels is to evaluate whether Akt activates beta-catenin by phosphorylation at Ser552 in the nephron*.

We have added this experiment but our current data cannot support the reviewers’ hypothesis; we find staining for this antibody in the medial and distal segment (Figure 6) but this doesn’t change upon inhibition of BMPR or PI3K.

*Writing/Text issues*:

*The number and organization of the figures, movies and supplementary figures interferes with the readability of the manuscript. Reducing the number of main and supplementary movies, figures and panels will enhance the legibility of the manuscript*.

We have tried to find a balance between reducing the figures and maintaining essential data. For instance, given the wide scepticism of specificity of pharmacological inhibitors we decided to keep all control experiments for this in figure supplements. Unfortunately, where in a biochemical paper this might be combined in a single western blot experiment, in our organ culture system this requires a lot more figures. On the other hand, in the old manuscript we showed the whole kidneys in figure supplements that our close-up data in the main figures was taken from to confirm we showed representative parts of the kidneys. To reduce complexity the majority of fields showing whole kidneys have now been removed and we have left the close-ups of nephrons.

*The presentation of the data and conclusions drawn needs to be more structured, and some of the statements, e.g. those of Reviewer 2, need to be more explicitly stated upfront (issues with interpreting the Lgr5 data, fact that Wnt KO's have no consequences thus far to nephron development, etc.). Reviewers 1-3 provided some specific recommendations for recrafting the text to more coherently flow for the reader*.

We have extended this; more reviewer-specific comments are addressed below.

*Please pay attention to Reviewer 3's comment that more is known regarding nephron development than the authors sometimes claim, as this is important to correct. Among them, the example cited is a good one, point out your claim that “The nephron β-catenin activity gradient is to our knowledge the first instance of a β-catenin activity gradient being described in any system”, when in fact this has been shown for multiple systems, including the hair follicle, hematopoietic and intestinal stem cell lineages*.

We thank the editor and reviewers’ for this comment and we have now toned down such statements or deleted them altogether where appropriate as this is not central to our work described here.

*The review comments below should put this summary into context*:

Reviewer #1:

*This is a very interesting manuscript that defines an important role for canonical WNT signaling in the patterning of the nephron. Some evidence is provided suggesting that there is normally a WNT gradient going from distal (strong) to proximal (weak). Increasing WNT, chemically or genetically, results in increased distal and decreased proximal regions of the nephron. Decreasing WNT gives more complicated results, although it is argued that there is a more rapid maturation of the proximal nephron*.

*The work is quite thorough in many respects, for example in using a multitude of chemical modifiers of WNT signaling, as well as genetic approaches. The characterization of resulting phenotypes is also generally excellent, with many outstanding images provided and a reasonable set of markers employed*.

We thank this reviewer for these kind and positive remarks.

*There are some significant unanswered questions that are raised by the results and the proposed model. As mentioned in the paper*, *which WNT is responsible for the proposed gradient? And why don't any Wnt knockouts show the predicted nephron patterning defect?*

We address this point in the response to the joint review.

*Further, why does knockdown of WNT in the proximal region of the nephron, where WNT is already proposed to be extremely low*, *partially rescue the Notch mutant loss of proximal structures?*

We hypothesize that under normal conditions proximal cells try to maintain a sufficiently low β-catenin activity level. By inhibiting this further using IWR1 we would simply be giving these cells a helping hand, and reaching the proximal fate becomes easier and is reached quicker. Our new Notch inhibition data shows more homogenous medium to high β-catenin activity levels throughout the nephron. By combining this with IWR1 or PI3K inhibition, which on their own decreases β-catenin activity throughout the nephron or in the medial segment (Figure 6), we would reduce β-catenin activities again to compensate for the Notch inhibition effects. However, we emphasize that we do not claim that inhibiting β-catenin is the only role of PI3K or Notch, making the above merely speculative although we added it to the extended discussion on the role of Notch.

*Reviewer #2*:

*The authors have investigated the role of canonical Wnt signaling in nephron patterning during murine renal development in ex vivo culture experiments. In addition, the effect of BMP and PTEN/PI3K/AKT signaling on nephron patterning as well as the interaction between these pathways was investigated*.

*Extensive intervention studies have been undertaken with both pharmacological compounds as well as mutant mouse models, and the effect on patterning is assessed morphologically, immunohistochemically and by gene expression analysis. The results provide relevant and new insight into mechanisms involved in nephron patterning*.

We thank the reviewer for these kind words.

*Major comments*:

*The number and organization of the figures and movies interferes with the readability of the manuscript. Reducing the number of movies, figures and panels will enhance the legibility of the manuscript*.

As discussed above and described in more detail below, we have addressed this in the best possible way, at least in our opinion.

*The presentation of the data and conclusion should be more structured. Conclusions are presented inconsistently in the manuscript body, the figure legends (*Figure 2—figure supplement 1*) and the figure itself (*Figure 6—figure supplement 1*). Paragraphs in the Results section sometimes end with a conclusion and sometimes with just research results. The manuscript contains redundant information and conclusions: e.g.: “Since proliferation was not affected by either increasing or decreasing β-catenin signaling” and “we found that proliferation along the nephron axis does not change in response to stimulating or inhibiting β-catenin”, both in the Discussion, and “phospho- and dephospho-specific antibodies (indicative of protein targeted for break-down and active 400 β-catenin respectively)” and “antibodies for β-catenin that was already tagged for degradation”, in the Discussion and in the Results respectively. Improving the cohesion (in the presentation) of the different experiments would strengthen the manuscript*.

We have, with the help of the reviewers’ comments, hopefully addressed some of the inconsistencies within the manuscript. We believe that most of these came from not explaining clearly enough our conclusions. We have now added short conclusions to the different paragraph which should be helpful to the reader. Some of the examples that the reviewer considered being redundant information and conclusions have been modified, but others we chose to keep as they have been touched open by other reviewers as important.

We have also taken into account all the reviewers’ and editor’s comments to improve the flow and cohesion of the manuscript.

We have had a critical look at which figure panels are essential, and as discussed above, we have extended the range of markers to be consistent throughout the manuscript in parts where there was some inconsistent use.

*Nephron patterning is assessed using several parameters: maturation, segment length, protein expression and gene expression. The results of these distinct assessments are not explicitly compared and integrated into one conclusion*.

We have added a section to the start of the Discussion combining these different types of data in a single model.

*A changing set of segment specific markers is used for nephron segment identification (e.g.*
Figures 4 and 6*)*.

*Although some of the pathway modulations are performed intermittently (canonical Wnt activation by CHIR, followed by CHIR withdrawal,*
Figure 2*), other pathway modulations are performed only continuously (e.g. BMP)*.

As discussed above, we now use a consistent set of markers and we have included a more general interpretation of the data. We have also added the reversal experiments for the other pathways as discussed above.

*Vague remarks should be replaced by more clearly defined statements. Different terms are used interchangeably in the Discussion section, but remain devoid of proper definition. Please clarify how these terms relate to the parameters measured (see above): “more efficient formation of the proximal domain”, “a strong positive effect on the development of the medial segment”, “increased rate of maturation”, “change in cell identity”*.

We now specifically state that we use the nomenclature used by the GenitorUrinary Development Molecular Anatomy Project (gudmap.org) for S-Shaped body nephrons to refer to distal, medial, and proximal segments. This is also mentioned in the text.

*In the Discussion section, the sentence “Exploring the roles… of future studies”: In the Results, the authors mention that Lgr-5 is a β-catenin target, however Lgr5 is also a Wnt receptor component that can enhance canonical Wnt signaling (*[28]*). The Lgr-5 expression (gradient) might therefore not be the result but the cause of the β-catenin gradient. The latter hypothesis is rejected by the experiments with Lgr-5 KO kidneys. This consideration should be mentioned more explicitly in the manuscript*.

This is a very helpful comment and it has been added to the manuscript.

*The authors state that*
Figure 1
*suggest that* β-*catenin activity is induced by (factors secreted by (?)) a single source with a first order decay*. *This concept is difficult to visualize due to the spiral morphology of the immature nephron. Please clarify. (One might however hypothesize that the* β-*catenin activity-inducing signal is propagated intraluminally.)*

As discussed in our response to reviewer 1, this is indeed something we cannot address within the scope of this manuscript. Moreover, tools for tracking Wnt molecules in the 3D context of a developing context would be as useful as they would be technically challenging (for multiple reasons). Transport within the lumen is indeed a possibility as we now discuss.

*β-catenin staining in*
Figure 1
*and Figure 1–figure supplement 2B, D only shows membrane bound β-catenin and not the nuclear β-catenin necessary for the transcription of Wnt target genes. In addition, the presence of phosphorylated β-catenin in the developing nephron in Figure F appears to be limited to one cell, which is insufficient to support the claim of a gradient of β-catenin phosphorylation made in the Discussion section. In addition, the supposed gradient of phosphorylated β-catenin along the tubule is insufficiently demonstrated by Figure 1–figure supplement 2B. These figures do not sufficiently explain the β-catenin activity gradient*.

This is addressed in the response to the joint review.

*Reviewer #3*:

*In this manuscript, the authors investigate pattern formation of the nephron. Although nephron development is a relatively neglected area of research, more is known regarding its development than the authors sometimes claim. Nonetheless, this manuscript provides a far more detailed and integrated view of nephron pattern formation than any other study to date. First, the authors conduct a detailed series of experiments showing that a beta-catenin activity gradient determines positional identity in a morphogenetic manner. This is shown primarily by using pharmacological inhibitors/activators of Wnt signaling along with a reporter of beta-catenin activity. The authors do a nice job verifying the specificity and fidelity of their system. Importantly, this pharmacological study is further supported genetically using various hypomorphic alleles of beta-catenin. The authors then show that, similar to the intestine, nephron patterning is further regulated through a BMP/PI3K network, which acts in modulating the more central-acting beta-catenin activity. Given the structural and functional similarities between the nephron and intestine, this is an attractive model. The authors also attempt to link their data with Notch signaling, which is also known to regulate nephron patterning. This section is somewhat confusing and is not well-integrated with the first two major findings. Overall, the authors have provided a more comprehensive study of nephron patterning than any previous study, and their study sheds some light on its subject that could not be gleaned from the current literature*.

We also thank this reviewer for the kind words.

*The results showing no changes in proliferation or apoptosis should be better explained and integrated into the manuscript. Do the authors suggest this implies that differentiation is mainly responsible for the patterning effects? Actually, changes in proliferation and apoptosis are well-known mechanisms for regulating morphogenesis (along with differentiation). At any rate, the authors should better clarify the purpose of these experiments and what they conclude from the results*.

We now explain this in more detail in the Discussion.

*The finding that the PI3K pathway is regulating beta-catenin is interesting. These pathways are known to interact, especially in the intestinal and hematopoietic systems, where Akt can activate beta-catenin by phosphorylation at Ser552. It would be interesting to see whether and where pS552-bcat positive cells were present during nephron patterning. Such data could further support the author's model*.

As discussed above, this experiment has been added in the new manuscript and has indeed strengthened our model. We thank the reviewer for this suggestion.

*The authors sometimes discount previous findings. For instance, they claim that “The nephron β-catenin activity gradient is to our knowledge the first instance of a β-catenin activity gradient being described in any system”. This is surprising since such gradients have been described in multiple systems, including mammals*.

This is addressed in the response to the joint review.

*The authors should consider either better integrating and explaining the Notch data or removing this section from the manuscript. It would be appropriate to discuss what's known about Notch and how it might relate to their new findings in the Discussion section; however, the current presentation tends to leave the reader more confused than enlightened*.

As discussed in the response to the joint review, we have added data to strengthen the links between Notch, PI3K and β-catenin and this has greatly improved the robustness of our proposed model.

We have also introduced new data showing 60x 1 µm scans through TCF/LEF nephrons treated with DAPT and in combination with Ly294002 and IWR1. These new data show how in DAPT treated nephrons the whole nephron is positive of GFP. It is not clear whether this is a cause or effect since DAPT treatment stops the proximal segments from forming. IWR1 treatment on top of DAPT treatment effectively reduces the signaling further and this might explain why some Wt1+ cells are now rescued. DAPT+ Ly29 does not clearly reduce GFP expression but it does rescue growth and expression of Jag1.